# Fibrocystin/Polyductin releases a C-terminal fragment that translocates into mitochondria and suppresses cystogenesis

Rebecca V Walker [1], Qin Yao[1,6], Hangxue Xu[1], Anthony Maranto [1], Kristen F Swaney[2], Sreekumar Ramachandran[2], Rong Li [2,3], Laura Cassina[4], Brian M Polster [5], Patricia Outeda[1], Alessandra Boletta [4], Terry Watnick[1] & Feng Qian [1] ✉

Fibrocystin/Polyductin (FPC), encoded by *PKHD1*, is associated with autosomal recessive polycystic kidney disease (ARPKD), yet its precise role in cystogenesis remains unclear. Here we show that FPC undergoes complex proteolytic processing in developing kidneys, generating three soluble C-terminal fragments (ICDs). Notably, ICD₁₅, contains a novel mitochondrial targeting sequence at its N-terminus, facilitating its translocation into mitochondria. This enhances mitochondrial respiration in renal epithelial cells, partially restoring impaired mitochondrial function caused by FPC loss. FPC inactivation leads to abnormal ultrastructural morphology of mitochondria in kidney tubules without cyst formation. Moreover, FPC inactivation significantly exacerbates renal cystogenesis and triggers severe pancreatic cystogenesis in a *Pkd1* mouse mutant *Pkd1^{V/V}* in which cleavage of *Pkd1*-encoded Polycystin-1 at the GPCR Proteolysis Site is blocked. Deleting ICD₁₅ enhances renal cystogenesis without inducing pancreatic cysts in *Pkd1^{V/V}* mice. These findings reveal a direct link between FPC and a mitochondrial pathway through ICD₁₅ cleavage, crucial for cystogenesis mechanisms.

Polycystic kidney disease (PKD) refers to a collection of genetic disorders characterized by the development of kidney cysts originating from the renal tubules, often accompanied by the growth of cysts in the hepatic and pancreatic ducts[1]. These cysts disrupt the renal parenchyma, leading to filtration obstruction and eventual kidney failure. PKD presents in two major forms, each with similar yet distinct pathological features: autosomal recessive polycystic kidney disease (ARPKD) and autosomal dominant polycystic kidney disease (ADPKD)[1,2].

ARPKD, affecting 1 in 20,000 live births, is typically diagnosed *in utero* or shortly after birth[3]. It is characterized by bilateral cystic enlargement of the kidneys and congenital hepatic fibrosis[4]. During early fetal development, the ARPKD kidney undergoes a transient phase of proximal tubule cyst formation, but as fetal stages progress, the site of cystic dilation shifts to distal tubules[5]. The typical renal pathology in postnatal specimens is non-obstructive fusiform dilatation of the collecting ducts. ARPKD is primarily caused by mutations in polycystic kidney and hepatic disease 1 (*PKHD1*) gene[6,7], although pathogenic variants in other genes have also been associated with ARPKD-like phenotypes[4,8]. The *PKHD1* gene produces a full-length transcript consisting of 67 exons that encode a protein called Fibrocystin/Polyductin (FPC) of 4074 amino acids (aa).

[1]Division of Nephrology, Department of Medicine, University of Maryland School of Medicine, Baltimore, MD, USA. [2]Department of Cell Biology, Johns Hopkins University School of Medicine, Baltimore, MD, USA. [3]Mechanobiology Institute and Department of Biological Sciences, National University of Singapore, Singapore 117411, Singapore. [4]Division of Genetics and Cell Biology, IRCCS San Raffaele Scientific Institute, Milan, Italy. [5]Department of Anesthesiology and Center for Shock, Trauma, and Anesthesiology Research, University of Maryland School of Medicine, Baltimore, MD, USA. [6]Present address: Laboratory of Clinical Investigation, National Institute on Aging, National Institutes of Health, Baltimore, MD, USA. ✉e-mail: fqian@som.umaryland.edu

The specific function of FPC and its role in cystogenesis are still unknown[6,7].

FPC is a receptor-like single transmembrane (TM) glycoprotein with a short cytoplasmic C-terminal tail that contains various motifs including a ciliary targeting sequence (CTS)[9] and nuclear translocation signal (NLS)[10,11]. It is primarily expressed in renal tubular structures, particularly in the collecting ducts and the thick ascending limb of the loop of Henle, with lower levels observed in the proximal tubules[12–15]. FPC has been reported to localize to primary cilia, and other subcellular compartments[12,13,15,16]. Studies using recombinant FPC[10,11] and epitope-tagged *Pkhd1* knock-in mouse models[14,17] have shown that a subset of FPC undergoes complex Notch-like proteolytic cleavage. Cleavage in the extracellular region at a putative proprotein convertase site KRKR$^{3613}$↓N (↓ denotes scissile bond) results in a soluble extracellular domain ("PECD") that remains associated with the TM-containing C-terminal fragment ("PTM") by disulfide bonds[11,14] or is secreted into urine[11,17]. Further intramembrane cleavage leads to the generation of one or multiple intracellular C-terminal domain fragments ("ICDs"), which may translocate into the nucleus and play a role in transcriptional regulation[10,11]. The exact processing pattern of native FPC and its functional implications are still unknown.

ADPKD is a more common condition, affecting 1 in 500–1000 individuals, and it typically manifests later in life[18]. This condition is characterized by the development of focal cysts originating from all segments of the renal tubules. The majority of ADPKD cases are caused by mutations in *PKD1* (-78%) or *PKD2* (-15%), which encode polycystin-1 (PC1) or polycystin-2 (PC2), respectively ("polycystins" collectively)[1,2]. PC1 is a 4302-aa atypical G-protein coupled receptor (GPCR) with 11 transmembrane domains, an extensive extracellular region, and a short cytoplasmic C-terminus. It plays an essential role in tubular morphogenesis[19]. *Pkd1* knockout mice develop severe cystic kidney and pancreas, culminating in embryonic lethality[20,21]. The protein is functionally regulated by *cis*-autoproteolytic cleavage at the juxtamembrane GPCR Proteolysis Site (GPS) within the GPCR-Autoproteolysis INducing (GAIN) domain[22–24]. *Pkd1$^{V/V}$* knock-in mice, which carry the T3041V mutation at the cleavage site of GPS, display impaired cleavage of the PC1 protein and develop PKD after birth[23,24]. PC1 can form a complex with PC2, a member of the transient receptor potential family. This complex is thought to function as a non-selective cation channel at the primary cilium and plasma membrane[25,26].

The regulation of mitochondrial ultrastructure is crucial for optimal mitochondrial function, affecting oxidative phosphorylation and overall metabolic capacity[27]. Dilated cristae have been linked to defective assembly of respiratory chain complexes in supercomplexes, resulting in reduced respiratory efficiency, impaired cellular metabolism, and decreased cell viability[28]. In ADPKD, metabolic reprogramming and mitochondrial dysfunction are significant contributors to disease pathogenesis[29,30], modifying renal cyst growth in *Pkd1* mouse models[31,32]. Polycystins were found to localize to the mitochondria-associated ER membranes[33] and potentially contribute to the regulation of $Ca^{2+}$ uptake in mitochondria[33–35]. PC1 may directly impact mitochondrial function and cystogenesis by translocating a small C-terminal cleavage product into the mitochondrial matrix[36,37] and indirectly by inhibiting miR-17 and modulating mitochondrial morphology[38]. Pathogenic *PKHD1* truncating mutations introduced into human embryonic kidney HEK293 cells led to significant structural and functional mitochondrial abnormalities[39]. Despite these findings, a consensus has not been reached concerning the role of mitochondria in PKD.

Various *Pkhd1* mutant mouse models have been developed to investigate the function of FPC and its role in ARPKD. However, these orthologous mouse mutants[15,40–42], including the *Pkhd1* knockout strain *Pkhd1$^{LSL/LSL}$*[17] (referred to as *Pkhd1 KO*) and the hypomorphic *Pkhd1$^{Δ3-4/Δ3-4}$* mutant carrying a deletion of exons 3–4[14,43] (Table 1), display minimal renal disease. Instead, these models develop varying degrees of biliary dysgenesis or pancreatic cysts in a subset of adult animals, but not during development[15,17,40,43]. Notably, the *Pkhd1$^{Δ67/Δ67}$* mutant strain, with a deletion of exon 67 encoding the last 137 aa of the 185-aa C-terminal tail, exhibits no extra-renal abnormalities[14]. The reason for the limited renal pathologies remains unclear. Investigation of *Pkhd1* mutations on a *Pkd1* mutant background reveals a genetic interaction between the two genes in cystogenesis[43,44]. In a recent study by Olson et al.[45], a digenic model combining the *Pkhd1 KO* strain with the hypomorphic *Pkd1$^{RC/RC}$* mutant displayed an ARPKD-like phenotype characterized by rapidly progressing PKD and early lethality. The study found no evidence of physical interaction between FPC and polycystins, suggesting independent functions that synergistically contribute to cystogenesis via a cilia-dependent mechanism[45].

In this study, we have investigated the intricate processing of endogenous FPC in developing kidneys and its impact on mitochondrial function and cyst formation in mice. We report that complex proteolytic processing of FPC generates three soluble C-terminal fragments. Among these fragments, ICD$_{15}$ was found to translocate into mitochondria through a mitochondrial targeting sequence located at its N-terminus. We investigated the functional consequences of FPC inactivation on mitochondrial respiration and analyzed the effects of ICD$_{15}$ translocation into mitochondria on mitochondrial function in cultured renal epithelial cells. We also analyzed the ultrastructural morphology of mitochondria in the *Pkhd1 KO* kidneys despite the absence of cystic phenotype. Through genetic epistasis analyses of three *Pkhd1* mouse mutants with the hypomorphic *Pkd1$^{V/V}$* mutation, we elucidated the role of FPC and the functional significance of its ICD$_{15}$ in cystogenesis during development. Our findings establish a direct connection between the ciliary FPC protein and mitochondria facilitated by ICD$_{15}$ through proteolytic cleavage, shedding new light on the function of FPC in cystogenesis in conjunction with PC1.

**Table 1 | Mouse mutant alleles and strains utilized in genetic epistasis analyses**

| Allele | Short-hand | Notes | Mouse details | Reference |
|---|---|---|---|---|
| *Pkhd1$^{LSL}$* | - or KO | *Pkhd1 KO* | https://www.jax.org/strain/019423 | Bakeberg et al., 2011 https://www.ncbi.nlm.nih.gov/pmc/articles/PMC3250208/[17] |
| *Pkhd1$^{Δ3-4}$* | Δ3–4 | Hypomorphic allele | https://www.pkd-rrc.org/downloads/pkhd1delta3-4/ | Garcia-Gonzalez et al., 2007 https://pubmed.ncbi.nlm.nih.gov/17575307/[43] |
| *Pkhd1$^{Δ67}$* | ΔCT | Deletes exon 67 that encodes C-terminal 137 aa. | https://www.pkd-rrc.org/downloads/pkhd1delta67/ | Outeda et al., 2017 https://www.ncbi.nlm.nih.gov/pmc/articles/PMC6005173/[14] |
| *Pkhd1$^{Flox67HA}$* | HA | A triple HA -tag is knocked-in at the C-terminus of FPC. | https://www.pkd-rrc.org/downloads/pkhd1ha/ | Outeda et al., 2017 https://www.ncbi.nlm.nih.gov/pmc/articles/PMC6005173/[14] |
| *Pkd1$^{V}$* | V | *Pkd1* GPS mutant, expressing uncleavable PC1$^{V}$. | https://www.pkd-rrc.org/downloads/pkd1v-2/ | Yu et al., 2007 https://pubmed.ncbi.nlm.nih.gov/18003909/[24] |

## Results

### Complex processing of native FPC resulting in C-terminal fragments in vivo

To investigate the role of proteolytic cleavage in native FPC, we examined the cleavage pattern of endogenous FPC in developing kidneys using a panel of anti-FPC antibodies through western blot analysis (Fig. 1a). The E1 antibody specifically recognizes an epitope within the C-terminal 137 amino acids encoded by the final exon, exon 67[14]. Additionally, we have developed two new rat monoclonal antibodies, E3 and E4, which target distinct epitopes within the extracellular region of mouse FPC. E3 is directed towards the PECD region, while E4 specifically targets the PTM region.

E1 antibody detected the ~500 kDa full-length FPC (indicated with "a" in Fig. 1b) as well as the ~50–55 kDa PTM product ("c") in P6 wild-type (WT) mouse kidney lysates, consistent with previous findings[11,14]. The antibody also recognized a novel product of ~190 kDa (indicated by "b"). Additionally, E1 detected three small fragments (indicated by "d", "e" and "f"). These products were not observed in kidney lysates from *Pkhd1 KO* mice, indicating that they are specific FPC products containing the C-terminal region. To confirm the observed FPC processing pattern, we utilized the *Pkhd1*[Flox67HA] knock-in model, which incorporates a 3xHA epitope tag at the C-terminus of FPC[14]. In the *Pkhd1*[Flox67HA] kidneys, the E1 antibody identified a matching set of six FPC products, with the four smaller products exhibiting slightly slower migration compared to the untagged FPC products detected in WT kidneys (Fig. 1b). These products were also specifically recognized by anti-HA antibody, but not in WT and KO samples, confirming the presence of the 3xHA tag (~4.4 kDa) and providing an explanation for the observed slower migration of the smaller products. This outcome validates the in vivo cleavage pattern of FPC in developing kidneys. Importantly, the presence of the C-terminal 3xHA tag enables us to further conclude that the three ICD fragments share the same C-terminal amino acid sequence but have different N-termini, likely resulting from distinct cleavage events.

To map the N-termini of these ICD fragments more accurately than was possible with a standard protein ladder, we designed a molecular ruler consisting of constructs beginning with methionine (Met) residues within the ICD (Fig. 1c, d). By using this ladder as a reference, the largest ICD fragment ("d", ~15.3 kDa, designated $ICD_{15}$) aligned approximately with the M0 or M1 construct beginning at position 3920 or 3924 respectively, the next fragment ("e", ~11.9 kDa, designated $ICD_{12}$) aligned closely with the MC construct starting at position 3953, and the smallest fragment was smaller than any of our ladder constructs ("f", ~6.3 kDa, designated $ICD_6$). Figure 1f schematizes all six FPC products, including the ICD products found in developing kidneys.

To gain insights into the functional role of these C-terminal products, we performed in silico analyses for this region of FPC. Using MitoProt II[46], we identified a potential mitochondrial targeting sequence (MTS) located at the N-terminus of $ICD_{15}$ (Fig. 1d). In mouse FPC (mFPC), the MTS spans amino acids Met[3924]-Leu[3952], while in human FPC (hFPC), it encompasses amino acids Met[3930]-Ala[3966], with probability scores of 0.3943 and 0.9799, respectively (Fig. 1e). The MTS partially overlaps with the nuclear localization signal (NLS) by seven amino acids. Notably, the sequence of $ICD_{15}$ is entirely encoded by exon 67 (Fig. 1d). The relationship between these targeting sequences and the three ICD products is schematized in Fig. 1f.

We next investigated the processing of endogenous FPC in the extracellular region using E3 and E4 antibodies (Fig. 1g). Both antibodies detected full-length FPC in WT and *Pkhd1*[+/−] kidney samples, similar to E1 antibody. However, no signal was observed in the *Pkhd1 KO* samples, confirming the specificity of these newly developed antibodies (Fig. 1g). However, E3 did not recognize any fragments shorter than full-length FPC that were detected by the E1 antibody. This suggests that once generated by cleavage, the PECD is not significantly

accumulated in the developing kidneys and may mostly be secreted in the urine[11,14,17]. On the other hand, the E4 antibody detected a small quantity of the shorter fragments that are also recognized by the E1 antibody (Fig. 1g, "b", "c"). These results suggest that in vivo, only a subset of native FPC molecules are processed at the putative proprotein convertase site, resulting in an abundance of the full-length form, a small amount of PTM product, and minimal accumulation of PECD.

The labeling pattern of E1, E3, and E4 antibodies was also investigated for FPC in kidneys derived from *Pkhd1*[Δ67/Δ67] (referred to as *ΔCT* hereafter), which lack exon 67[14]. As anticipated, E1 did not detect any FPC products in the mutant kidneys (Fig. 1g). In contrast, both E3 and E4 antibodies detected a mutant form of FPC in *ΔCT* kidneys, which was similar in size to, but less abundant than, the full-length FPC present in WT and *Pkhd1 +/−* samples (Fig. 1g). Our data indicate that the *ΔCT* mutant expresses a truncated FPC protein, FPC-ΔCT, which lacks the $ICD_{15}$ region but retains the intact extracellular region, transmembrane domain, and adjacent ciliary targeting sequence. The functional significance of the $ICD_{15}$ region will be described later in the study.

### FPC $ICD_{15}$ translocates to mitochondria through an unmasked N-terminal mitochondrial targeting sequence

Given that classical MTSs are typically located at the N-terminus of proteins[47], we hypothesized that the predicted MTS in full-length FPC might be masked or inaccessible. We further hypothesized that cleavage events leading to the generation of $ICD_{15}$ could expose the MTS at the N-terminus, thereby facilitating the mitochondrial import of this cleavage product. To test this hypothesis, we generated human and mouse constructs of $ICD_{15}$ tagged with GFP at the C-terminus and assessed their localization upon transfection in mIMCD3 (murine inner medullary collecting duct) cells (Fig. 2a–d). Both the human (Fig. 2b, h$ICD_{15}$-GFP) and mouse (Fig. 2c, m$ICD_{15}$-GFP) constructs localized to mitochondria, along with some weak nuclear and cytoplasmic staining. In contrast, when a mutant mouse $ICD_{15}$ construct (m$ICD_{15}$-Mut-GFP) with a mutation in the MTS region that reduces the probability of mitochondrial import was examined (Fig. 2a), it displayed a complete loss of mitochondrial accumulation and instead exhibited strong nuclear localization (Fig. 2d). This observation strongly suggests that the N-terminal MTS of $ICD_{15}$ is crucial for its translocation into mitochondria. Notably, the human $ICD_{15}$-GFP construct demonstrated a more robust mitochondrial accumulation compared to the mouse construct, which appeared more diffused. This is consistent with the higher probability score for mitochondrial import in the human MTS sequence (0.9799) compared to the corresponding mouse sequence (0.3943) (Fig. 1e).

We then used a split GFP complementation assay[48] to validate the mitochondrial import of $ICD_{15}$ (Fig. 2e–j). In this assay, the first 10 β-strands of GFP ($GFP_{1-10}$) were fused with mCherry and targeted to mitochondria through a mitochondria-targeting sequence (MTS-mCherry-$GFP_{1-10}$), while the 11th β-strand ($GFP_{11}$) was fused to the $ICD_{15}$ ($ICD_{15}$-$GFP_{11}$) for assessment (Fig. 2e). Mitochondrial GFP fluorescence would only occur if $ICD_{15}$-$GFP_{11}$ is translocated into mitochondria and spontaneously assembles with the MTS-mCherry-$GFP_{1-10}$. When mouse $ICD_{15}$-$GFP_{11}$ (m$ICD_{15}$-$GFP_{11}$) and MTS-mCherry-$GFP_{1-10}$ were co-transfected into mIMCD3 cells, we observed strong green fluorescence complementation specifically in mCherry-labeled mitochondria (Fig. 2f). Similarly, the human $ICD_{15}$-$GFP_{11}$ construct demonstrated robust reconstitution of GFP fluorescence within mitochondria (Fig. 2g). Co-transfection with the $GFP_{11}$ construct lacking the $ICD_{15}$ portion did not produce green fluorescence, despite the observation of mCherry fluorescence within the mitochondria (Fig. 2j).

To further confirm the role of the MTS in $ICD_{15}$ for mitochondrial translocation, we introduced specific mutations in the MTS region of the human $ICD_{15}$-$GFP_{11}$ construct (hICD15-$GFP_{11}$), designed to decrease the probability of mitochondrial import, as predicted by MitoProt II.

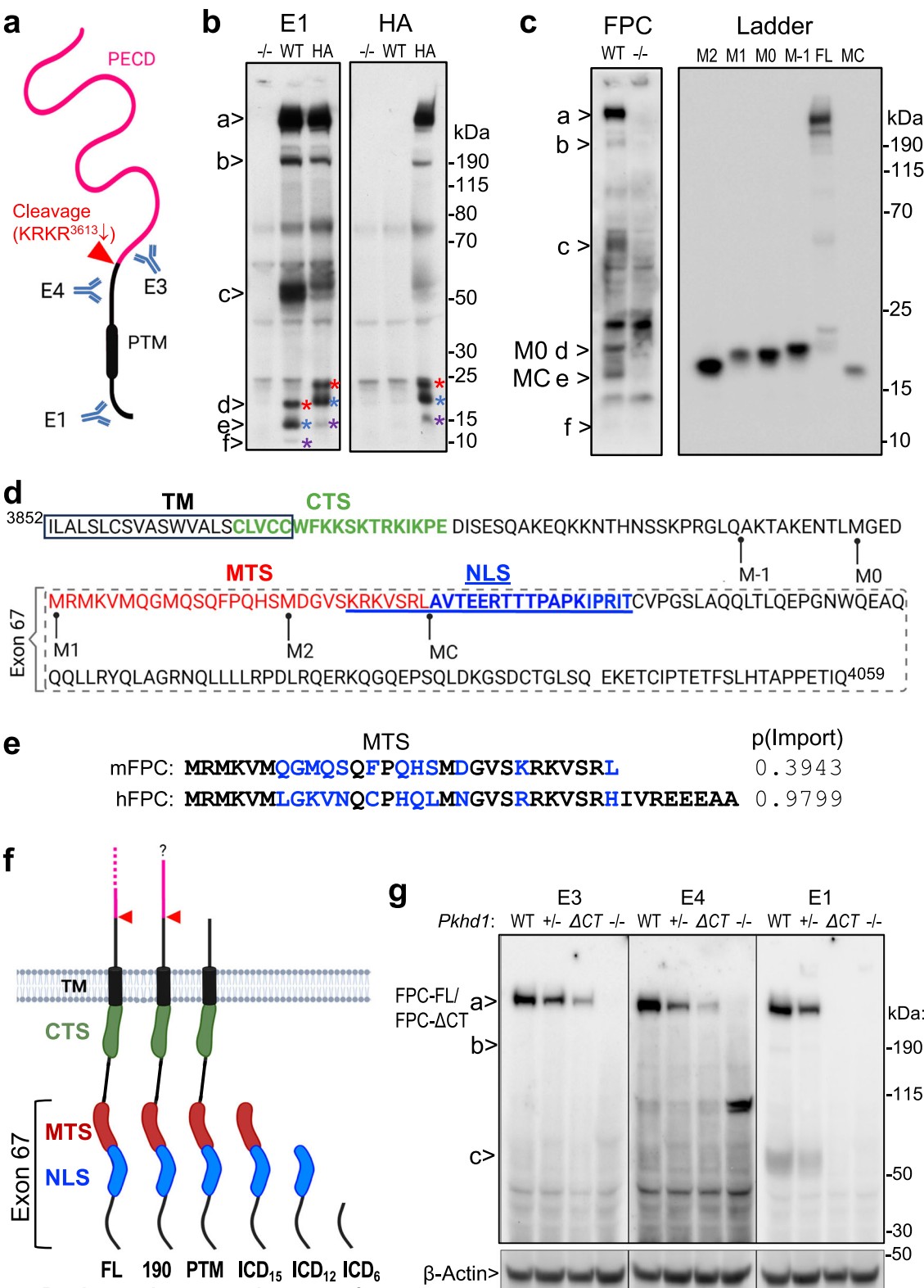

Substituting Arg[3931] and K[3933] with Glu (Mut1), which decreased the probability of mitochondria import from 0.9799 to 0.0428 (Fig. 2a), resulted in reduced GFP fluorescence complementation as evidenced by weakened mitochondrial GFP signal (Fig. 2h). Additional replacement of Met[3935] and Val[3939] with Glu (Mut2), which further reduces the import probability to 0.0226, completely abolished the mitochondrial GFP signal (Fig. 2i). A weak diffuse cytosolic GFP signal was occasionally observed for the MTS mutants, possibly because they interacted with the newly synthesized GFP[1-10] construct outside of mitochondria. Collectively, these findings provide compelling evidence for the essential role of the MTS in mediating the translocation of ICD[15] into mitochondria.

We next characterized the ICD[12] and ICD[6], which lack the MTS. For this purpose, we generated stable MDCK (Madin-Darby canine kidney) cell lines expressing the transmembrane C-terminus of mouse FPC (TMCT, aa 3852–4059, see Fig. 1d) to allow posttranslational cleavage.

**Fig. 1 | Processing of native FPC generates C-terminal cleavage fragments in vivo, including ICD$_{15}$ with an N-terminal mitochondrial targeting sequence. a** Diagram depicting the epitope positions of the anti-FPC antibodies, with PECD (magenta) and PTM (black) indicated. Figure created in BioRender. **b** Western blot of *Pkhd1 KO*, WT and HA P6 mouse kidneys using E1 (left panel) and anti-HA (right panel) antibodies. Various FPC products are indicated by ("a"–"f"), with corresponding ICD products marked by color-matched asterisks. Three independent experiments were performed, with similar results. **c** Western blot with E1. Six bands ("a"–"f") are detected from WT (+) kidney lysates but not in lysates from knockout (-): The full-length ("a"), novel -190 kDa band ("b"), PTM ("c"), and three small C-terminal bands at -15 kDa ("d"), -12 kDa ("e"), and 6 kDa ("f") are indicated. A ladder of various low molecular weight constructs (MC to M-1) and the full-length FPC construct (FL) is used to map the size of the endogenous bands; their start sites are indicated in (**d**). Three separate experiments were performed, with similar

results. **d** Functional motifs in mouse FPC C-terminal region. The predicted mitochondrial translocation sequence (MTS) is in red. The previously identified nuclear localization sequence (NLS)[10] is blue and underlined. The transmembrane domain (TM) is enclosed in a box. The ciliary targeting sequence (CTS)[9] is in green. The dashed box indicates the ICD$_{15}$ region encoded by exon 67. The ladder construct start points are indicated for M-1 to MC for the ladder used in (**c**). **e** Sequence alignment of human and mouse FPC MTS. Non-identical amino acids are colored blue. p(Import): the probability of mitochondrial import. **f** Diagram of predicted FPC fragments indicating the position of the functional motifs and the Exon 67 encoded region, along with the labels of the respective bands. The predicted cleavage site in the extracellular domain is indicated. Figure created in BioRender. **g** Western blot of WT, +/-, *ΔCT*, and *KO* (-/-) P6 kidneys using E3, E4 and E1 antibodies. Loading control: β-Actin. Two separate experiments were performed, with similar results. Source data are provided as a Source Data file.

We performed a cellular fractionation to enrich for mitochondria and detected bands corresponding to ICD$_{15}$, ICD$_{12}$, and ICD$_6$ in the mitochondrial fraction, but not in the cytoplasmic fraction (Fig. 2k). These bands were absent from the control cells. This result suggests that ICD$_{12}$ and ICD$_6$ are likely derived from ICD$_{15}$ within the mitochondria.

### Role of FPC in mitochondrial function and the functional impact of ICD$_{15}$ in cultured renal epithelial cells

To investigate the potential function of FPC and the role of ICD$_{15}$ in mitochondria, we inactivated *Pkhd1* in mIMCD3 cells by employing CRISPR technology and evaluated the effect on mitochondrial function (Fig. 3). We observed that the *Pkhd1* knockout (KO) cells displayed a significant reduction in oxygen consumption rate (OCR) compared to the wild-type control (CT) mIMCD3 cells, as determined by a standard Mito Stress Test conducted on a Seahorse flux analyzer (Fig. 3a). This result indicates that the inactivation of FPC directly contributes to significant alterations in mitochondrial function, thus suggesting a crucial role of FPC in the regulation of mitochondrial function.

To assess the functional significance of ICD$_{15}$ on mitochondria, we stably expressed mouse ICD$_{15}$ constructs (ICD$_{15}$ or ICD15-Mut) in mIMCD3 *Pkhd1* KO and CT cells using the PiggyBac transposon system[25,49] (Fig. 3b) and evaluated their impact on OCR (Fig. 3c). Remarkably, the expression of ICD$_{15}$ in *Pkhd1* KO cells resulted in a modest but significant increase in maximal respiration and spare respiratory capacity when compared to *Pkhd1* KO cells. Conversely, the expression of ICD$_{15}$-Mut, which has reduced mitochondrial translocation (Fig. 2a, d), had a minimal or marginal impact on these parameters (Fig. 3c). These findings indicate that ICD$_{15}$ can enhance mitochondrial respiration through mitochondrial translocation, partially compensating for the impairment of mitochondrial function caused by *Pkhd1* deficiency.

Significantly, the expression of the ICD$_{15}$ construct also increased OCR in the CT cells, both in terms of maximal respiration and spare respiratory capacity, compared to the CT cells (Fig. 3c). Notably, the increase observed as a result of the mitochondrial translocation of the ICD$_{15}$ construct was more pronounced in CT cells than in *Pkhd1* KO cells. In contrast, minimal to no effects on OCR were observed with the ICD$_{15}$-Mut construct when compared to CT cells. These findings strongly suggest that ICD$_{15}$ plays a crucial role in enhancing mitochondrial function in renal epithelial cells, and its functional impact is dependent on its translocation into the mitochondria.

### Altered mitochondrial ultrastructure in *Pkhd1* KO mice

We examined mitochondrial ultrastructural morphology in the renal tubules of *Pkhd1 KO* and WT littermate controls using transmission electron microscopy (Fig. 4a–d). Focusing on PTs known for their high ATP demands and susceptibility to mitochondrial dysfunction[50], we imaged three P6 kidneys per genotype. A total of 962 mitochondria were compared for each genotype, from 3–5 proximal tubules per kidney. Mitochondrial morphology was evaluated using the following

established mitochondrial shape descriptors[51]: surface area, perimeter, circularity [4π(surface area/perimeter$^2$)], and solidity [(area/convex area)]. Circularity indicates the degree of resemblance to a perfect circle (with a value of 1), while solidity measures the concavity or convexity of a shape, reflecting its complexity and branching. Measurements were obtained from images blinded from genotype.

Despite the absence of a cystic phenotype, we observed significant differences in mitochondrial size between *Pkhd1 KO* samples and their WT littermate controls. *KO* mitochondria exhibited a smaller surface area ($p = 0.0080$) (Fig. 4e) and perimeter ($p = 0.0002$) (Fig. 4f) compared to WT littermate mitochondria. Additionally, *KO* mitochondria displayed a more rounded shape ($p = 0.0062$) (Fig. 4g) compared to WT mitochondria. However, there was no significant difference in branching, as measured by solidity, between WT and *KO* mitochondria ($p = 0.9643$) (Fig. 4h). These structural alterations suggest a higher degree of mitochondrial fragmentation in *KO* kidney tubular cells.

Furthermore, aberrantly shaped mitochondria in *KO* proximal tubules (Fig. 4j, l) exhibited distinct changes in cristae diameters compared to WT mitochondria (Fig. 4i, k). Analysis of 50 mitochondria from each group showed that *KO* mitochondria exhibited significantly swollen cristae, with an average diameter of 20.24 nm, compared to the WT average of 19.06 nm ($p = 0.0191$) (Fig. 4m). This finding suggests an impairment in the inner mitochondrial membrane in *KO* mitochondria. Taken together, our results provide compelling evidence suggesting that FPC likely plays a role in regulating both the structure and function of mitochondria in the tubular cells of the kidney.

### Disruption of FPC enhances the cystic kidney phenotype of non-cleavable *Pkd1* mutant, *Pkd1$^{V/V}$*

To investigate the role of FPC and the functional significance of the ICD$_{15}$ in cystogenesis, we conducted genetic epistasis analysis by combining *Pkhd1* mutant strains with the hypomorphic *Pkd1$^{V/V}$* knock-in mouse mutant[24] (Table 1). The *Pkd1$^V$* allele, which expresses a non-cleavable PC1$^V$ mutant, was specifically chosen for its unique effects on cystogenesis in specific nephron segments and developmental stages. This allele has been shown to rescue cystogenesis in the embryonic kidney and pancreas, preventing embryonic lethality[24]. Moreover, it provides protection to proximal tubules (PTs), while allowing cystic dilation in distal nephron segments and collecting ducts (CDs) postnatally, ultimately leading to kidney failure and death around 3 weeks of age[24]. We hypothesized that these distinctive characteristics exhibited by the *Pkd1$^V$* allele would significantly facilitate the identification of the specific contributions of FPC and the impact of ICD$_{15}$ on cystogenesis.

We initially examined the phenotypic impact of the hypomorphic *Pkhd1$^{Δ3-4}$* allele on the *Pkd1$^{V/V}$* mutants by comparing the phenotypes of the *Pkhd1$^{Δ3-4/Δ3-4}$* (*Δ3–4* hereafter) or *Pkd1$^{V/V}$* single

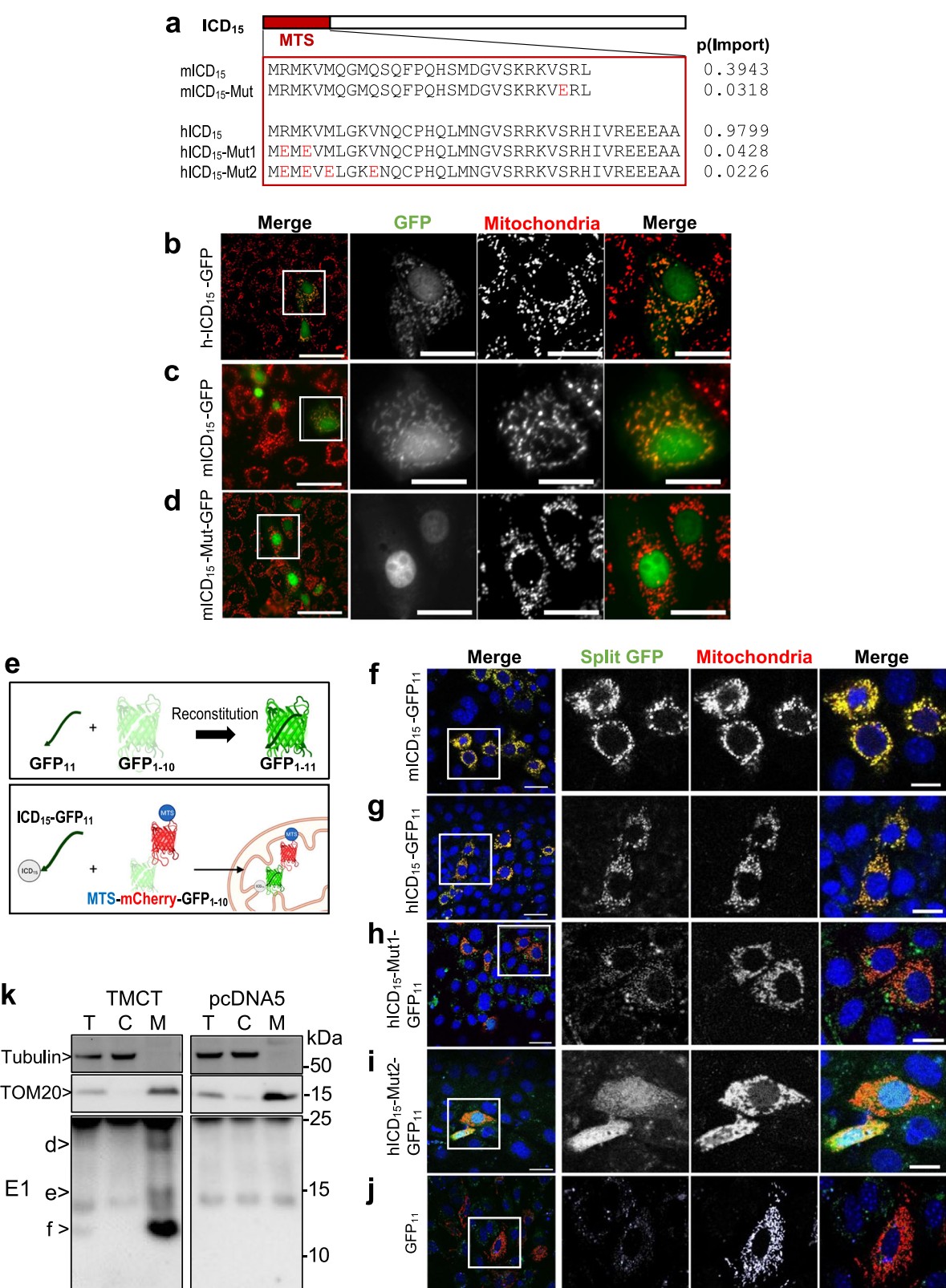

mutants with the digenic mutant mice. Histologically, at postnatal day 0 (P0), the kidneys of *Δ3−4* mutants (Fig. 5b) appeared similar to their WT littermates' kidneys (Fig. 5a), as confirmed by lectin staining (Fig. 5f, *Δ3−4*; Fig. 5e, WT). The *Pkd1*[V/V] kidneys exhibited intact proximal tubules (marked by *Lotus tetragonolobus* lectin, LTL, green) and displayed slight cystic dilation in the CDs (marked by *Dolichos biflorus* agglutinin, DBA, red) at this stage (Fig. 5d, h). Both digenic

*Pkhd1*[Δ3-4/+];*Pkd1*[V/+] heterozygous mutants and *Pkhd1*[Δ3-4/Δ3-4];*Pkd1*[V/+] trans-mutants were fertile and appeared indistinguishable from the WT animals. Therefore, we performed a series of intercrosses either between the digenic heterozygous mutants (*Pkhd1*[Δ3-4/+];*Pkd1*[V/+]) or between the trans-mutants (*Pkhd1*[Δ3-4/Δ3-4];*Pkd1*[V/+]). Breeding resulted in a total of 93 pups from 16 litters with only three digenic homozygous animals (*Pkhd1*[Δ3-4/Δ3-4];*Pkd1*[V/V], denoted as *Δ3−4/V* hereafter) surviving

**Fig. 2 | Mitochondrial Translocation of FPC ICD₁₅. a** Schematic diagram of mouse and human ICD$_{15}$ constructs used in the study, with the MTS sequences and the predicted p(Import) to mitochondria shown. The mutant amino acids are shown in red. **b–d** FPC ICD$_{15}$ constructs tagged with GFP were transfected into mIMCD3 cells and imaged. Mitochondria are shown in red, and FPC-GFP constructs in green. A field of cells is shown in the first panel, scale bar 50 μm. Box highlights area of magnified cells. Scale bar 20 μm. Three separate experiments per construct were performed independently with similar results. **e** Schematic of the split-GFP β-barrel complementation (upper panel). FPC ICD$_{15}$ fragment reconstitutes GFP barrel if mitochondria localization is achieved (lower panel). Figure created in BioRender. **f–j** Images of GFP$_{11}$-fused FPC ICD$_{15}$ fragments and GFP$_{1-10}$-fused mitochondria-localized mCherry in mIMCD3 cells. Leftmost image shows wide view merge, scale bar 10 μm. The three rightmost panels (GFP, mCherry, Merge) show a magnified image of the box in the first panel, scale bar 5 μm. Mouse and human ICD$_{15}$ split GFP constructs used are indicated on the left. Panel **j** shows a GFP$_{11}$-only Control. Three separate experiments per construct were performed independently with similar results. **k** Western blot of MDCK cell fractions from cells expressing mouse FPC C-terminal construct TMCT, spanning amino acids 3852–4059, with HA and V5 tags at the N- and C-termini, respectively. T total protein, C cytoplasmic fraction, M mitochondrial fraction. FPC was detected using E1 antibody. MDCK cells containing pcDNA5 vector were used as a negative control. Tubulin and TOM20 are loading controls. ICD$_{15}$ ("d"), ICD$_{12}$ ("e"), and ICD$_6$ ("f") are indicated. kDa sizes correspond to less accurate commercial protein ladder. Three independent experiments per construct were performed with similar results. Source data are provided as a Source Data file.

to birth. Histologic examination of these rare *Δ3−4/V* pups revealed extensive PT and CD cystic dilation throughout the entire kidney, with little parenchyma remaining (Fig. 5i, m). We observed occasional glomerular cystic expansion in the *Δ3−4/V* mutants (indicated by the green arrow), which was not observed in any of the single mutants. Next, we examined the embryonic stages for gross pathology. The *Δ3−4/V* embryos that survived to embryonic day 17.5 (E17.5) showed extensive renal cystic dilation in both PT and CD and occasional glomerular cysts (Fig. 5j, n). *Δ3−4/V* kidneys exhibited cystic dilation in both PT and CD at E16.5 (Supplementary Fig. 1a, b) with slight dilations beginning to appear at E15.5 (Supplementary Fig. 1d). *Pkd1$^{V/V}$* mice did not exhibit any abnormalities in the PT and showed negligible dilations in the CD starting around E16.5 (Supplementary Fig. 1g). These findings show that the combined *Pkhd1$^{Δ3-4/Δ3-4}$* and *Pkd1$^{V/V}$* mutations result in a more severe kidney phenotype reminiscent of *Pkd1* null mutants[20,52].

To test the possibility of *Pkhd1* having an additional role in cystogenesis that may not have been fully uncovered with the hypomorphic *Pkhd1$^{Δ3-4}$* allele, we examined the phenotypic effects of the *Pkhd$^{LSL}$ KO* allele[17] on the *Pkd1$^{V/V}$* mutants. The *Pkhd$^{LSL/LSL}$* (*KO*) single mutants did not show any significant kidney abnormalities as expected (Fig. 5c, g). We performed a series of intercrosses similar to those performed with the *Pkhd1$^{Δ3-4}$* allele. Breeding resulted in a total of 136 pups from 19 litters with only 6 digenic homozygous *Pkhd1$^{LSL/LSL}$;Pkd1$^{V/V}$* mutants (referred to as *KO/V*) surviving to birth. The digenic *KO/V* mutants developed severely cystic kidneys at postnatal day 0 (P0) (Fig. 5k, o), similar to the *Δ3−4/V* mice (Fig. 5i, m). Cyst development was evident at the embryonic stages of E17.5 (Fig. 5l, p) and E16.5 (Supplementary Fig. 1j). PT and CD cysts appeared in all stages and occasional glomerular cysts were also noted (Fig. 5k, l, green arrows). Therefore, like the *Δ3−4/V*, the *KO/V* mutants display a severe *Pkd1* null-like cystic kidney phenotype[20,52].

To understand the basis for the similar phenotype of the *Δ3−4/V* and the *KO/V* kidneys, we compared FPC protein expression in *KO*, *Δ3−4*, and WT kidneys. Using the E1 antibody, we detected a ~500 kDa mutant FPC in *Pkhd1$^{Δ3-4/Δ3-4}$* kidneys that is similar in size to, but far less abundant than, the full-length FPC in WT sample, but not in the *KO* sample (Fig. 5q)[14]. The mutant FPC likely contains a deletion of 123 amino acids in the extracellular domain, presumably encoded by a minor mutant transcript previously identified in *Pkhd1$^{Δ3-4/Δ3-4}$* kidneys[43]. These findings indicate that *Pkhd1$^{Δ3-4}$* represents a severely hypomorphic allele, which provides an explanation for the observed phenotypic similarities observed between the *Δ3−4/V* and *KO/V* mutants.

In summary, the loss of FPC in hypomorphic *Pkd1$^{V/V}$* mice results in a kidney phenotype similar to the *Pkd1* null condition, with cystic expansion in glomeruli, proximal tubules, and accelerated cyst growth in distal tubules and collecting ducts. These results reveal the protective role of *Pkhd1* in maintaining the architecture of both the proximal tubule and distal tubule/collecting duct lumens during kidney development.

## Expression level and intracellular trafficking of mutant PC1 are not altered by FPC inactivation in developing kidneys

PC1 plays a central role in cyst formation in a genetic interaction network for polycystic kidney and liver diseases[44,53]. Proper functioning of PC1 requires cleavage at the GPS, resulting in PC1$_{NTF}$ and PC1$_{CTF}$ products that remain non-covalently associated, coexisting with a small amount of the uncleaved form, PC1$^U$[22,24,54]. To elucidate the mechanisms underlying the genetic interaction between *Pkhd1* and *Pkd1* in kidney cystogenesis, we examined whether the loss of FPC alters the expression and cleavage of PC1. To address this, we performed western blot analysis on P6 kidneys from *Pkhd1 KO* and WT littermates using the E8 antibody[55,56] (Fig. 5r). The E8 antibody specifically recognizes a region in PC1$_{CTF}$, thereby detecting both PC1$^U$ and PC1$_{CTF}$ on the western blot. This feature enables the assessment of the overall level of PC1 and the extent of cleavage by analyzing the signal ratio of PC1$_{CTF}$ to PC1$^U$ forms[56]. We found that the levels and ratios of PC1$_{CTF}$ and PC1$^U$ were comparable between the *Pkhd1 KO* and WT kidneys (Fig. 5r, left panel "Kidney"), consistent with the findings reported by Olson et al[45]. To ascertain that the effects of *Pkhd1* inactivation were specific to renal tubules and not confounded by PC1 expression outside of renal tubules of the kidney, we also isolated intact renal tubule fragments and analyzed them separately. We found that the levels and ratios of PC1$_{CTF}$ and PC1$^U$ were also similar in the isolated renal tubules from *Pkhd1 KO* and WT kidneys (Fig. 5r, left panel "Tubule"). The renal tubule preparation was validated by imaging and a set of nephron fragment-specific markers (Fig. 5r, right panel). These data indicate that *Pkhd1* inactivation does not affect the level and GPS cleavage of PC1 in kidney tubules.

To determine if the non-cleavable PC1$^V$ mutant is more susceptible to *Pkhd1* inactivation, we compared the level of PC1$^V$ in the kidneys of *Pkd1$^{V/+}$* and *Pkhd1$^{Δ3-4/Δ3-4}$;Pkd1$^{V/+}$* littermates using western blot analyses. These kidneys were chosen specifically because they exhibit a structurally normal phenotype, allowing for an accurate assessment of the effect on PC1$^V$ without the confounding factors of cyst development. Using the E8 antibody, we detected two distinguishable bands of over 500 kDa in size in both samples, with the lower band comigrating with the less abundant PC1$^U$, which is exclusively detected in the WT control (Fig. 5s). To differentiate between PC1$^V$ and PC1$^U$ in these samples, we took advantage of their differential glycosylation patterns. PC1$^V$ acquires partial resistance to Endoglycosidase H (EndoH) digestion, while PC1$^U$ remains entirely EndoH-sensitive[22,54]. Our *N*-glycan analysis using EndoH and PNGase F confirmed that the upper band in the two samples exclusively corresponded to the EndoH-resistant form of PC1$^V$ (PC1$^V$-R), while the lower band primarily represented the EndoH-sensitive form of PC1$^V$ (PC1$^V$-S), with a minor contribution of PC1$^U$ (Fig. 5s). We found similar levels of both PC1$^V$-R and PC1$^V$-S between the *Pkd1$^{V/+}$* and *Pkhd1$^{Δ3-4/Δ3-4}$;Pkd1$^{V/+}$* kidney samples (Fig. 5s). These results suggest that the inactivation of *Pkhd1* does not significantly affect the expression level or intracellular trafficking of PC1$^V$ in developing kidneys. Our findings indicate that the *Pkd1* null-like phenotype observed in the digenic mutants is not attributed to a

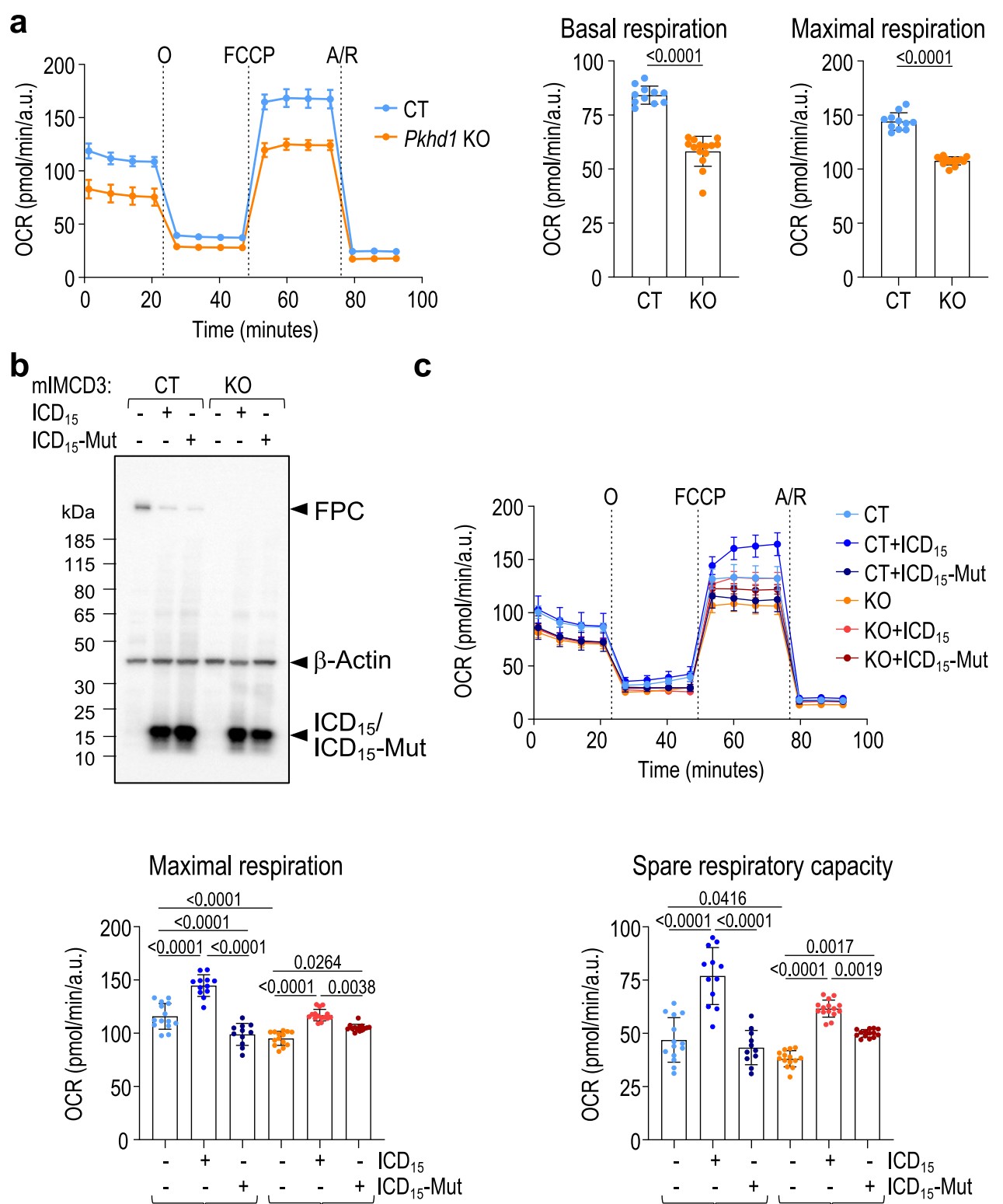

reduction in the level or maturation of PC1$^V$. Other factors or mechanisms may contribute to the observed phenotype in the digenic mutants.

**Deletion of *Pkhd1* exon 67 enhances renal cystogenesis of *Pkd1*$^{V/V}$ mutant mice but does not induce pancreatic cyst formation as observed with *Pkhd1* inactivation**

We next investigated the potential role of mitochondria-associated ICD$_{15}$ in cystogenesis by performing an epistatic analysis. For this

purpose, we utilized the *Pkhd1*$^{\Delta67/\Delta67}$ mutant strain, which specifically lacks the ICD$_{15}$ amino acid sequence (Table 1; Fig. 1g), in combination with the *Pkd1*$^{V/V}$ mutant strain. We performed intercrosses between digenic *Pkhd1*$^{\Delta67/+}$;*Pkd1*$^{V/+}$ heterozygotes or between the *Pkhd1*$^{\Delta67/\Delta67}$;*Pkd1*$^{V/+}$ trans-mutants to generate the digenic homozygous animals *Pkhd1*$^{\Delta67/\Delta67}$;*Pkd1*$^{V/V}$ (denoted as *ΔCT/V*). Breeding resulted in a total of 192 pups from 28 litters with 26 *ΔCT/V* animals surviving to birth, corresponding roughly to Mendelian ratios. We found that, unlike the neonatal lethal *KO/V* and *Δ3−4/V* mutants, the *ΔCT/V* survived an

**Fig. 3 | Expression of ICD$_{15}$ enhances mitochondrial function through mitochondrial translocation in cultured renal epithelial cells. a** Left: analysis of oxygen consumption rate (OCR) measurement of a representative experiment in *Pkhd1* KO and wild-type control (CT) mIMCD3 clones in basal condition and after sequential addition of oligomycin (O), FCCP and antimycin A/rotenone (A/R). Right: quantification of basal and maximal respiration of OCR measurements in left. Data are mean ± standard deviation of technical replicates (CT, $n = 11$ and KO, $n = 14$), statistical analysis: Two-tailed *t*-test, ****$p < 0.0001$. Results are representative of three independent experiments, each performed in $n = 10–16$ technical replicates. **b** The Western blot analysis using E1 revealed comparable expression levels of mouse ICD$_{15}$ and ICD$_{15}$-Mut constructs in both CT and *Pkhd1* KO mIMCD3 cells. Loading control: β-Actin. **c** Analysis of OCR measurement of a representative experiment in CT and *Pkhd1* KO mIMCD3 cells with or without expression of mouse

ICD$_{15}$ or ICD$_{15}$-Mut as indicated, in basal condition and after sequential addition of the mitochondrial inhibitors as indicated. Lower two panels: Quantification of maximal respiration (left) and spare respiratory capacity (right) of OCR measurements. Data are mean ± standard deviation of technical replicates (from left to right, CT, $n = 14$, 12 and 11, KO, $n = 14$, 15 and 14); statistical analysis: one-way ANOVA. Maximal respiration, *$p = 0.0264$, **$p = 0.0038$, ****$p < 0.0001$. Spare respiratory capacity, *$p = 0.0416$, **$p = 0.0017$ (KO, -ICD$_{15}$ *vs* +ICD$_{15}$-Mut), **$p = 0.0019$ (KO, +ICD$_{15}$ *vs* +ICD$_{15}$-Mut), ****$p < 0.0001$. Results for control cells are representative of five out of five independent experiments, and results for *Pkhd1* KO cells are representative of three out of five independent experiments, each performed in $n = 10–16$ technical replicates. Source data are provided as a Source Data file.

average of 20 days postnatally. We examined 13 *ΔCT/V* kidneys at P0. Significantly, whilst *ΔCT* single homozygote kidneys were cyst-free (Fig. 6b, e), resembling WT kidneys (Fig. 6a, d), *ΔCT/V* mice had developed numerous discernible renal cysts (Fig. 6c, f), though to a lesser extent than *KO/V* and *Δ3–4/V* embryos. The cystic area of *ΔCT/V* kidneys was significantly greater than that of WT and *ΔCT*, which were indistinguishable by this measurement, or that of *Pkd1^{V/V}* (V) (Fig. 6g). Significantly, like in *KO/V* and *Δ3–4/V* embryos, lectin staining revealed that these cysts also developed from proximal tubule and collecting duct origins (Fig. 6f). However, glomerular cysts were uncommon in the *ΔCT/V* mutants, reflecting either the reduced severity of the cystic state or a lack of importance of ICD$_{15}$ in maintaining the Bowman's capsular space. At P6, the *ΔCT/V* mice showed cystic dilation extended to the entire kidney similar to the *Pkd1^{V/V}*, but also significant PT dilation that is absent in the *Pkd1^{V/V}* single mutants (Supplementary Fig. 1k, l). Taken together, our data provide strong evidence supporting the critical role of the mitochondria-associated ICD$_{15}$ product in regulating renal cystogenesis, in conjunction with PC1, during development.

Lastly, we examined the role of FPC and the impact of ICD$_{15}$ in cystogenesis within the liver and pancreas by analyzing their phenotypes of the *Δ3–4/V*, *KO/V*, and *ΔCT/V* digenic mutants during development. In the liver, we did not observe any apparent abnormalities in either the digenic or single mutants (Supplementary Fig. 1a, c, d–f, i, j). This contrasts with the findings reported in non-orthologous murine models of ARPKD[57]. However, due to perinatal death, we were unable to examine hepatic fibrosis at a later stage. In the pancreas, both *Δ3–4/V* and *KO/V* mutants exhibited extreme cystic expansion, leading to the formation of a single central cystic lumen and the obliteration of the parenchyma at E17.5 (Fig. 6m, n). This cystic pancreatic phenotype was also observed in the digenic mutants as early as E15.5 (Supplementary Fig. 1d-f), with only a small amount of remaining parenchyma at the periphery. This phenotype closely resembles that observed in the *Pkd1* mutant embryos[20]. However, the pancreas in the *ΔCT/V* mice remained intact at E17.5 (Fig. 6o), appearing similar to the WT (Fig. 6h) and the single mutants (Fig. 6i–l; Supplementary Fig. 1h,i). Our findings demonstrate that, while FPC plays a crucial role in cystogenesis in the pancreas, its ICD$_{15}$ does not have a substantial impact on this process when combined with the *Pkd1^{V/V}* mutation.

## Discussion

In our study, we made significant discoveries regarding the role of FPC and its C-terminal cleavage product ICD$_{15}$ in mitochondria and cystogenesis. We found that ICD$_{15}$ contains a mitochondrial targeting sequence (MTS) at its N-terminus (Fig. 1), which directs its translocation into mitochondria (Fig. 2). ICD$_{15}$ enhances mitochondrial respiration in cultured renal epithelial cells through mitochondrial translocation and partially restores the impaired mitochondrial function resulting from the loss of FPC (Fig. 3). Furthermore, we observed significant ultrastructural abnormalities in the

mitochondria of *Pkhd1* knockout kidney tubules despite the absence of a cystic phenotype (Fig. 4). Moreover, we found that the loss of FPC in hypomorphic *Pkd1^{V/V}* mouse mutants resulted in a severe cystic phenotype in the kidney and pancreas during development, closely resembling the phenotype of *Pkd1* null mice (Figs. 5, 6). Specific deletion of exon 67, which encodes the ICD$_{15}$ amino acid sequence, enhances cystogenesis in the kidney but not in the pancreas of *Pkd1^{V/V}* mice (Fig. 6), supporting a significant role of ICD$_{15}$ specifically in kidney cyst formation. Overall, our findings establish a crucial connection between FPC and mitochondria, likely mediated by ICD$_{15}$ through proteolytic cleavage. This mechanism likely plays a crucial role in protecting the proper morphology of renal tubular structures and in cystogenesis in conjunction with PC1.

Our findings, which reveal altered mitochondrial ultrastructure and function in the *Pkhd1* knockout kidney and cultured cells, indicate that FPC plays a critical role in regulating mitochondrial structure and function. Furthermore, our analysis of ICD$_{15}$ in mitochondrial respiration provides compelling evidence that this cleavage product is involved in mediating the mitochondrial function of FPC. By targeting ICD$_{15}$ to the mitochondria, it may participate in the regulation of essential mitochondrial processes, such as respiration and the production of critical metabolites involved in cell metabolism. Notably, our study revealed that ICD$_{15}$ expression has a more pronounced effect on mitochondrial respiration in wild-type control mIMCD3 cells than in *Pkhd1* KO cells. This observation suggests that the functional impact of ICD$_{15}$ on mitochondria may be affected by the full-length FPC or other FPC cleavage products. These FPC forms contain the ciliary targeting sequence and may exert their function at primary cilia; the full-length FPC may mostly reside in the endoplasmic reticulum[14]. They may affect the responsiveness of mitochondria to ICD$_{15}$, leading to a more specific effect of ICD$_{15}$ on mitochondrial functions via unknown mechanisms. These observations suggest a complex interplay between FPC, ICD$_{15}$, and mitochondrial function.

The genetic interaction between *Pkhd1 KO* or *Pkhd1^{Δ3-4}* allele and the *Pkd1^{V}* allele leads to a phenotype similar to the *Pkd1* null phenotype in the kidney and pancreas, two major organs affected in ADPKD. This finding suggests that FPC acts in conjunction with PC1, potentially by modulating the specific function of PC1 or interacting with downstream components of the PC1-specific pathway to maintain the proper morphology of renal tubular structures. A previous study found no evidence of physical interaction between full-length forms of FPC and polycystins in mouse kidneys[45]. However, a recent study identified a soluble C-terminal cleavage product of PC1 (CTT) that undergoes translocation to the mitochondria matrix[36], impacting mitochondrial morphology, function, and cystogenesis[36,37]. It is plausible that FPC and PC1 may synergistically regulate a specific mitochondrial function through their respective cleavage products within mitochondria. It remains to be determined how their cleavage products are generated, and whether they physically interact with each other or with other molecules within mitochondria to regulate mitochondrial function.

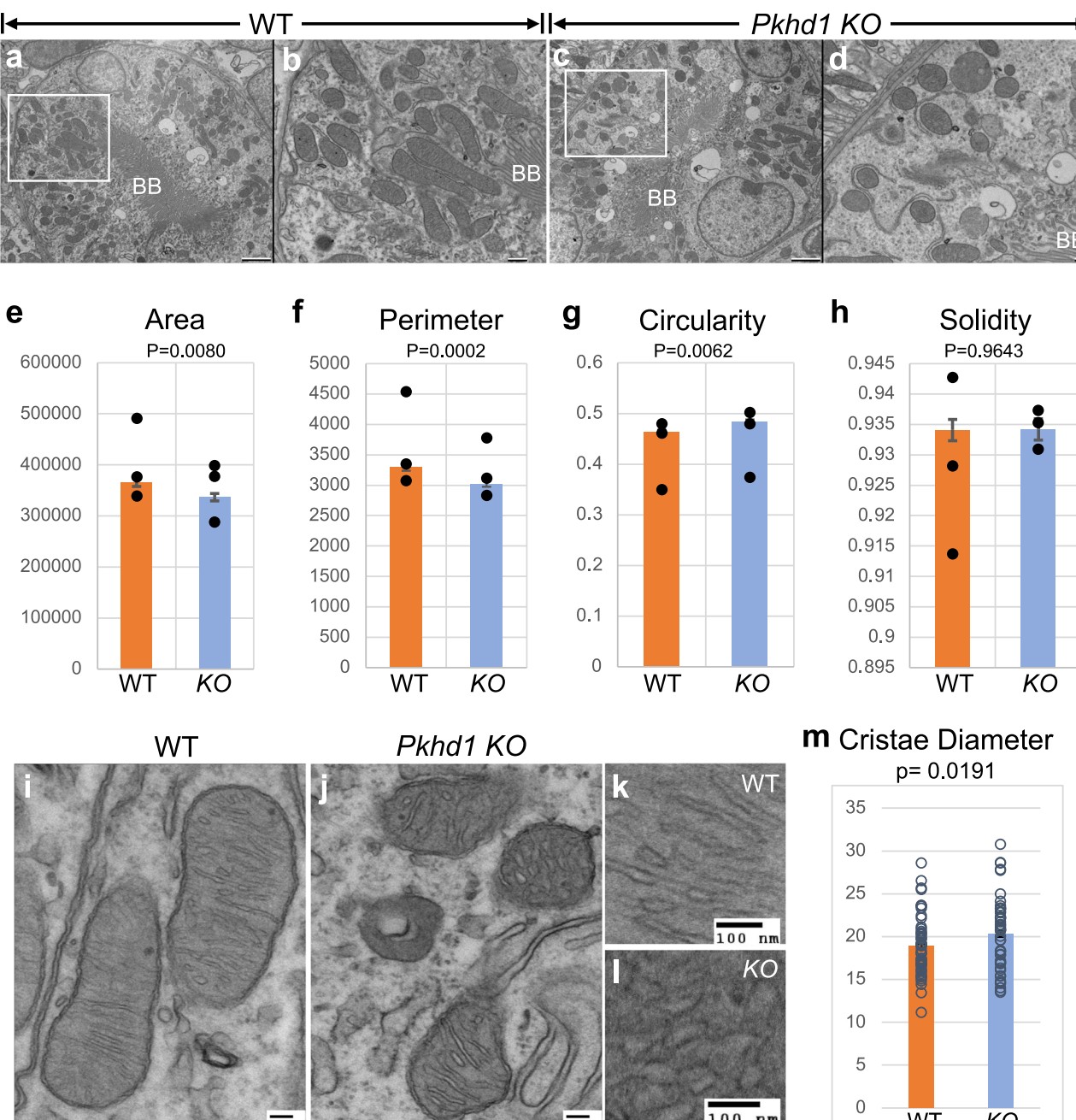

**Fig. 4 | Ultrastructural alteration of mitochondria of *Pkhd1 KO* kidney tubules.**
**a–d** Electron micrographs of WT and *Pkhd1 KO* kidney proximal tubules. **a** WT proximal tubule identified by brush border (BB) at the center of the lumen. Scale bar 2 µm. The area outlined in the white box is magnified in (**b**). **b** Magnified area of WT image. Scale bar 500 nm. **c** *KO* proximal tubule identified by brush border (BB) at the center of the lumen. Scale bar 2 µm. The area outlined in the white box is magnified in (**d**). **d** Magnified area of *KO* image. Scale bar 500 nm. **e–h** Graphs depicting shape descriptors. **e** Area, (**f**) Perimeter, (**g**) Circularity, and (**h**) Solidity. A total of 962 mitochondria were measured from 3 kidneys, using images from 3–5 proximal tubules per kidney. Points on each graph represent the average for each

kidney measured and the bar represents the average of all mitochondria measured. Error bars represent the standard error of the mean (SEM). *P*-values were derived by Student's *t*-test. **i–l** Electron micrographs of mitochondria in WT and *Pkhd1 KO* kidney proximal tubules. **i** WT mitochondria. Scale bar 100 nm. **k** Magnification of WT cristae. Scale bar 100 nm. **j** *KO* mitochondria. Scale bar 100 nm. **l** Magnification of *KO* cristae. Scale bar 100 nm. **m** Graph shows average cristae diameter in nm. Each point represents the average diameter of cristae in a single mitochondrion. 50 mitochondria measured per genotype. Error bars represent SEM. *P*-values were derived by Student's *t*-test. Source data are provided as a Source Data file.

The effects of *Pkhd1* inactivation on mitochondrial structure and function observed in our study exhibit both similarities and differences when compared to a previous study conducted on *PKHD1* mutant HEK293 cells[39]. While both studies observed cristae dilatation, there were discrepancies, such as larger mitochondria and increased OCR in the HEK293 cells. These differences may stem from variances in cell types, *PKHD1/Pkhd1* mutations, and PC1 expression levels,

underscoring the complex nature of FPC's involvement in mitochondrial regulation.

One key finding highlighting the critical role of FPC in cystogenesis is the profound impact of *Pkhd1^{Δ3-4/Δ3-4}* or *KO* mutations on kidney and pancreas cystogenesis in hypomorphic *Pkd1^{V/V}* mice, resulting in a phenotype similar to *Pkd1* null. While PC1 has been established as a central player in cyst formation within the genetic interaction network

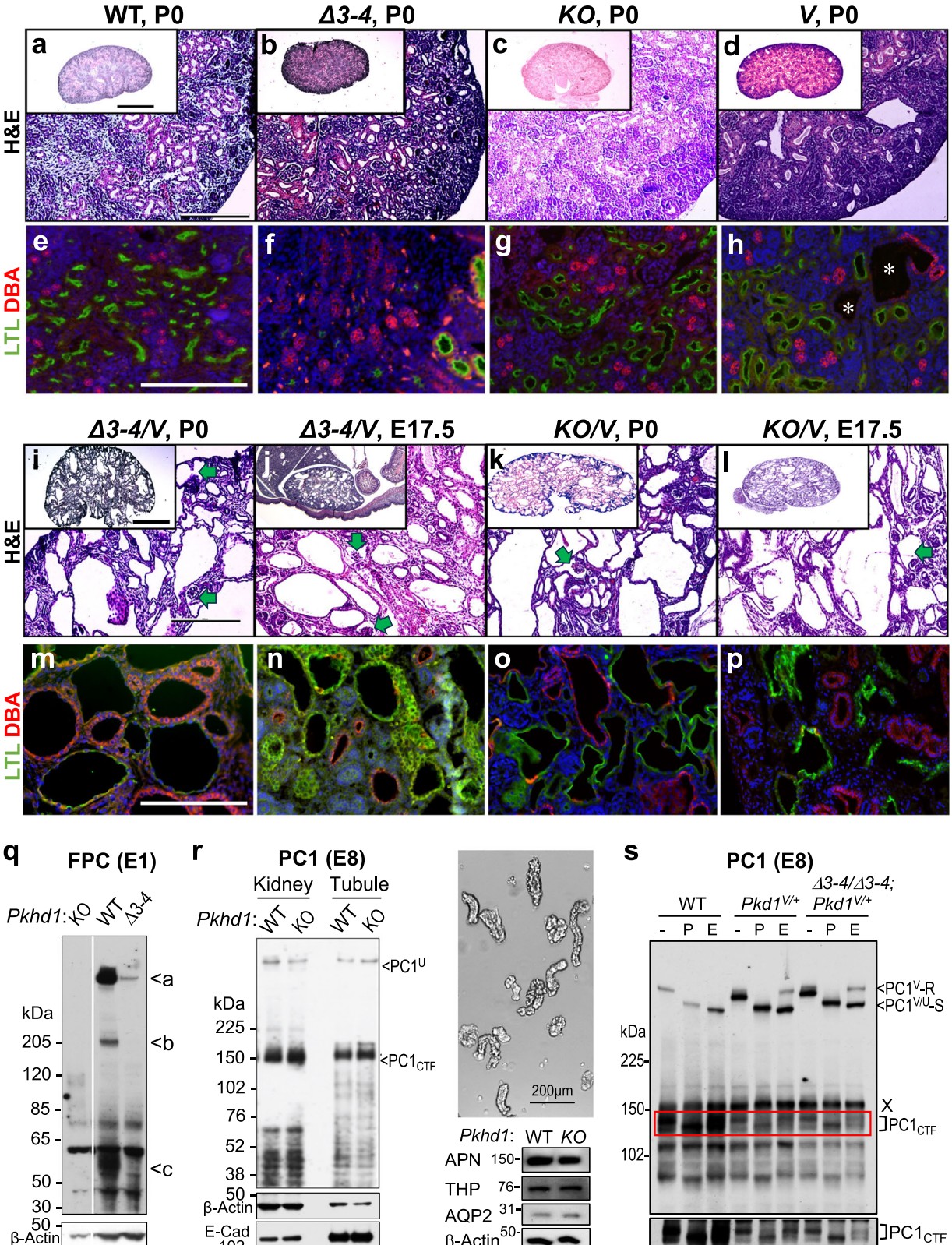

for polycystic kidney and liver diseases[44,53], the observed *Pkd1* null-like phenotype in the digenic mutants cannot be explained by reduced levels or maturation of PC1$^V$. Instead, our epistasis analysis using the *Pkhd1*$^{Δ67/Δ67}$ and *Pkd1*$^{V/V}$ mutant strains suggests the involvement of the mitochondria-associated ICD$_{15}$ specifically in the kidney cyst development. Interestingly, the presence of the *Pkhd1*$^{Δ67/Δ67}$ mutation in *Pkd1*$^{V/V}$ mutants results in a less severe cystic phenotype. This

observation could be attributed to the expression of FPC-ΔCT protein, which retains the intact ciliary targeting sequence and is likely directed to the cilia. This ciliary function may counteract renal cystogenesis through additional mechanisms involving its ecto- and transmembrane domains (See Fig. 7 for further discussion). In the pancreas, this ciliary function may be fully sufficient to protect the organ from cystogenesis through mechanisms beyond its C-terminal tail.

**Fig. 5 | *Pkhd1* mutations enhance the *Pkd1^{V/V}* cystic kidney phenotype without altering mutant PC1^V.** **a–d** Hematoxylin and eosin (H&E) staining of representative P0 kidney sections of WT and single mutant genotypes. **a** WT, (**b**) *Δ3–4* (*Pkhd1^{Δ3-4/Δ3-4}*), (**c**) *KO* (*Pkhd1^{LSL/LSL}*), (**d**)*V* (*Pkd1^{V/V}*). Scale bar 500 μm. Inset: whole kidney slice, scale bar 1 mm. **e–h** Lectin staining of representative P0 kidney tubule sections of the corresponding genotypes. White asterisks in **h** indicate cysts. Proximal tubule marked by *Lotus tetragonolobus* lectin (LTL)-green, distal tubule/collecting duct marked by *Dolichos biflorus* agglutinin (DBA)-red. Scale bar 200 μm. **i–l** H&E staining of representative kidney sections of digenic homozygote kidneys. **i** *Δ3–4/V* at P0,(**j**) *Δ3–4/V* at E17.5,(**k**)*KO/V* at P0,(**l**)*KO/V* at E17.5. Scale bar 500 μm. Glomerular cysts are indicated by green arrows. Inset whole kidney slice, scale bar 1 mm. **m–p** Lectin staining of kidney tubules of the corresponding genotypes. Scale bar 200 μm. 5–10 embryos were examined per genotype. **q** Western blot of P2 *Pkhd1 KO*, WT, and *Δ3–4* kidney lysates using E1. Loading control: β-Actin. **r** Western blot of WT and *Pkhd1 KO* kidney lysates and isolated tubule fragments (P6) using anti-PC1 antibody E8 (left panel). Loading controls: β-actin and E-cadherin. Right panel shows a graph of isolated renal tubule samples, scale bar 200 μm. The tubule samples were validated by western blots with indicated nephron segment markers: Aminopeptidase N (APN), proximal tubules; Tamm-Horsfall glycoprotein/Uromodulin (THP), thick ascending limb; aquaporin 2 (AQP2), collecting duct. **s** Western blot using E8 on WT, *Pkd1^{V/+}*, and *Pkhd1^{Δ3-4/Δ3-4};Pkd1^{V/+}* (*Δ3–4/Δ3–4;Pkd1^{V/+}*) P2 kidney isolates after incubation with buffer only (-), PNGase (P) or EndoH (E). Equal amounts of protein were loaded. PC1^V-R indicates the EndoH-resistant PC1^V. PC1^{U/V}-S indicates EndoH-sensitive PC1^U or PC1^V. PC1_{CTF} is detected in all three samples. Red box around CTF is shown at a higher exposure below. Note: an endogenous band at -160 kD cross-reacts with E8 as previously shown[54,66], indicated by X (this can be used as a loading control). All experiments were conducted with 2-3 repetitions with consistent results. Source data are provided as a Source Data file.

Previous studies have reported structural and functional alterations in mitochondria within the cystic kidney epithelia of *Pkd1* mutants, with implications for modifying renal cyst growth[31,32,36]. However, the precise relationship between these mitochondrial alterations, polycystins, and their role in cystogenesis remains unclear. Our study reveals mitochondrial abnormalities in *Pkhd1 KO* kidney tubules despite the absence of cyst formation. Based on our findings, we propose that *Pkhd1* inactivation creates a pro-cystic metabolic state due to the absence of FPC's ICDs, which in turn exacerbates the *Pkd1*-dependent cystogenic pathway that is associated with the CTT of PC1. The combined effects contribute to the increased severity of the cystic phenotype in digenic mutants. Our findings suggest that ICD_{15} may serve as a crucial link to the *Pkd1*-dependent cystogenic pathway, implying that mitochondrial impairment may play a more proximate role in cystogenesis in the kidney. Future studies are necessary to determine the mechanisms by which dysfunctional C-terminal cleavage products of both FPC and PC1 within mitochondria lead to mitochondrial abnormalities involved in cystogenesis.

The importance of ICD products in the pathogenesis of ARPKD is underscored by the identification of truncating mutations and missense variants specifically situated in exon 67 of the *PKHD1* gene in severe ARPKD patients[58–60]. For example, a frameshift mutation (V3925fs) combined with an F1785L mutation on the second allele was found in a severely affected ARPKD patient with enlarged cystic kidneys, chronic kidney disease, and portal hypertension[59]. The V3925fs frameshift mutation is predicted to result in the deletion of ICD_{15} by truncating FPC before the MTS. Another severely affected patient carried a variant P3968Lfs along with a C1431Y mutation on the second allele, resulting in truncation and loss of ICD_{12} due to disruption of the MTS cleavage site at P3968.

Our study suggests intriguing differences between human and mouse ICD_{15} in terms of mitochondrial import probability and accumulation, with human ICD_{15} displaying a higher import probability score and stronger mitochondrial accumulation than its mouse counterpart. These findings suggest that FPC, particularly through its ICD_{15} component, may exert a more significant role in mitochondrial function in humans, thereby implying that the loss of FPC in humans could have a more pronounced impact on mitochondrial function. This notion has important implications, especially considering the crucial involvement of mitochondrial disturbances in cyst formation during human fetal development. Evidence indicates that mutations in genes encoding mitochondrial proteins can independently result in cystogenesis in the kidneys of human fetuses[61,62]. Furthermore, human kidneys naturally have lower levels of functional PC1 due to abnormal alternative splicing of the *PKD1* gene, resulting in premature termination of PC1[63]. This intrinsic reduction in functional PC1, combined with the heightened impact of FPC on mitochondrial function, may contribute to the more severe pathology observed in ARPKD compared to mice.

Primary cilia play a role in inhibiting cyst formation in renal tubules but promote cyst growth in the absence of functional polycystins. Ma et al[64] have proposed a cilia-dependent cyst activation (CDCA) signal that is activated to drive cystogenesis when polycystins are lost. Under normal conditions, polycystins are thought to tonically repress the CDCA signal, maintaining kidney homeostasis and facilitating tubular adaptation to environmental cues.

We propose a molecular mechanism to explain the interaction between *Pkhd1* and *Pkd1* in cystogenesis through a cilia-mitochondria connection (Fig. 7). According to our model, FPC present in primary cilia may act as a sensor for the extracellular environment and transmit information to mitochondria through the release of its ICD_{15} via proteolytic cleavage. When cilia lack functional PC1, a CDCA signal is generated and propagated to mitochondria, amplified by the absence of PC1's CTT. FPC acts as a counterbalance to this signal both in cilia and through its mitochondrial ICD_{15} component. In *Pkhd1 KO* kidney, loss of FPC and its ICD_{15} alone has minimal effect on cystogenesis, at least in mice, due to the tonic suppression of CDCA by the presence of PC1. In *Pkd1^{V/V}* kidney, PC1^V's impaired ciliary trafficking[22,23] activates the CDCA signal, which may be further suppressed to some extent by PC1^V outside the ciliary compartment. FPC's ICDs further repress this signal within mitochondria, resulting in delayed and restricted renal cystogenesis[24]. In *ΔCT/V* kidney, the production of ICD_{15} is disrupted, preventing its ability to suppress the CDCA signal within mitochondria. However, FPC-ΔCT is still capable of localizing to the cilium and reducing CDCA levels. As a result, enhanced cystogenesis is observed compared to *Pkd1^{V/V}* single mutants. In the *Δ3–4/V* and *KO/V* kidneys, the concurrent loss of FPC eliminates its regulatory influence on the CDCA signal, including its effect within the mitochondria. As a consequence, unopposed CDCA activity ensues, leading to severe cystogenesis that closely resembles the cystic phenotype observed in *Pkd1* null mice. Further research is required to shed light on the specific molecular events and signaling pathways involved in the interplay between FPC and PC1 for the connection between cilia and mitochondria in polycystic kidney diseases.

## Methods
### Mice
Animal studies were performed in adherence to the NIH Guide for the Care and Use of Laboratory Animals and approved by the University of Maryland School of Medicine Institutional Animal Care and Use Committee (Protocol # 0421008). The mice were housed at a controlled temperature of 72 °F (±2 °F) with humidity maintained between 35% and 55%. They followed a 12-h light-dark cycle and had unrestricted access to LabDiet 5010-Laboratory Autoclavable Rodent Diet. Both male and female mice were used for experiments since phenotypes were consistent between the sexes. Littermates were compared when possible. All mice were congenic on a C57BL/6 J background. Embryos and neonates younger than 7 days of age were euthanized by decapitation using sharp scissors without prior sedation or anesthesia. Mice

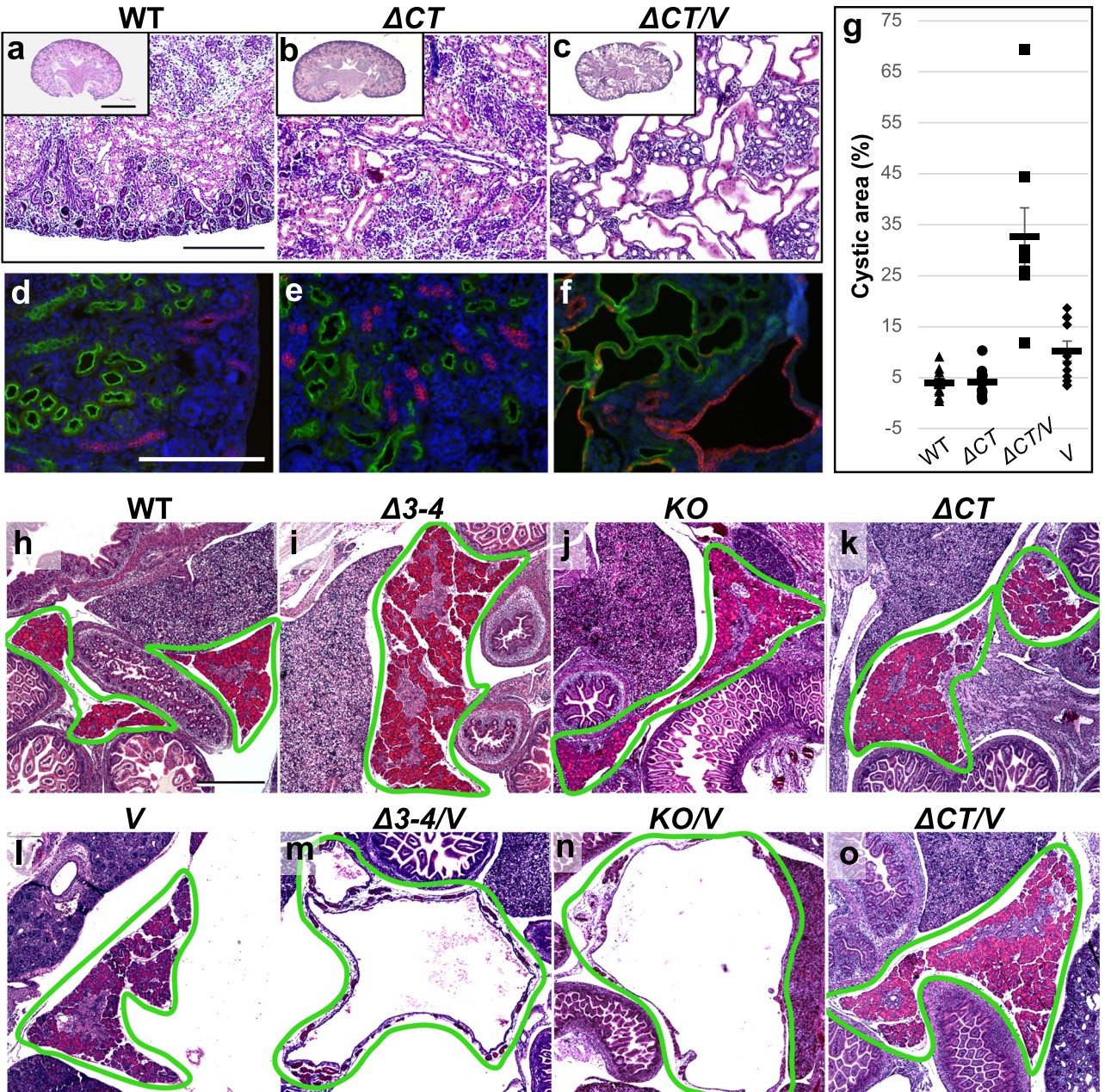

**Fig. 6 | The impact of ICD$_{15}$ deletion on the cystic phenotype of the kidney and pancreas in *Pkd1$^{V/V}$* mice. a–c** Hematoxylin and eosin (H&E) of representative P0 kidney sections, scale bar 500 μm. Inset: whole kidney slice, scale bar 1 mm. **a** WT, (**b**) *ΔCT* (*Pkhd1$^{Δ67/Δ67}$*), (**c**) *ΔCT/V* (*Pkhd1$^{Δ67/Δ67}$;Pkd1$^{V/V}$*). **d–f** Lectin staining of kidney tubules;(**d**) WT, (**e**) *ΔCT*, (**f**) *ΔCT/V*, scale bar 200 μm. Proximal tubule marked by *Lotus tetragonolobus* lectin (LTL)-green, distal tubule/collecting duct marked by *Dolichos biflorus* agglutinin (DBA)-red. **g** Graph representing cystic area in %. WT *n* = 10 mice, *ΔCT n* = 13 mice, *ΔCT/V n* = 8 mice, *V* (*Pkd1$^{V/V}$*) *n* = 10 mice. Error bars represent SEM. **h–o** H&E staining of representative pancreas sections at E17.5, scale bar 500 μm. Pancreas is outlined with a green line. Of note, loss of FPC, but not deletion of ICD$_{15}$, results in massive cystic dilation of the pancreas in *Pkd1$^{V/V}$* mutant mice. **h** WT, (**i**)*Δ3–4* (*Pkhd1$^{Δ3-4/Δ3-4}$*), (**j**)*KO* (*Pkhd1$^{LSL/LSL}$*), (**k**) *ΔCT* (*Pkhd1$^{Δ67/Δ67}$*), (**l**) *V* (*Pkd1$^{V/V}$*), (**m**) *Δ3–4/V* (*Pkhd1$^{Δ3-4/Δ3-4}$;Pkd1$^{V/V}$*) cystic pancreas, (**n**) *KO/V* (*Pkhd1$^{LSL/LSL}$;Pkd1$^{V/V}$*) cystic pancreas, (**o**) *ΔCT/V* (*Pkhd1$^{Δ67/Δ67}$;Pkd1$^{V/V}$*) non-cystic pancreas. At least 5 animals per genotype were independently examined with consistent results. Source data are provided as a Source Data file.

older than 7 days were euthanized using Isoflurane for anesthesia, followed by cervical dislocation to ensure euthanasia. Table 1 details the mice used in this investigation.

## Antibodies
### Homemade primary antibodies
**Mouse FPC**. We generated two new rat monoclonal antibodies, E3 and E4, directed to either side of the putative proprotein convertase site KRKR$^{3613}$↓N in the extracellular region of mouse FPC[11]. E3 is directed to

PECD N-terminal to the site (mouse FPC aa 3254-3353), while E4 is directed to PTM C-terminal to the position (mouse FPC aa 3610–3800). E1 is directed to the FPC C-terminus and has been previously described[14]. PC1: E8, a rat monoclonal antibody directed to CTF generated using a mouse polycystin-1 fragment (aa 3682–3882) as immunogen, has been previously described[54]. These antibodies are available through the Polycystic Kidney Disease Research Resource Consortium (https://www.pkd-rrc.org/fibrocystin-antibody/) and were used for immunoblotting at used at 1:500–1000 dilution.

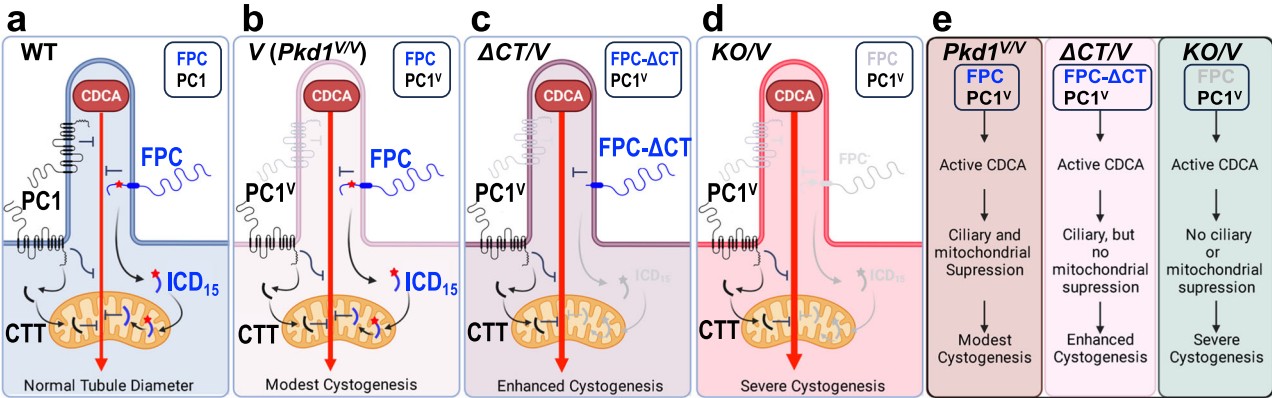

**Fig. 7 | Model of FPC and Polycystin-1 interaction in the pathogenesis of polycystic kidney disease via a cilia-mitochondria connection. a** In WT kidney, the ciliary component of the cilia-dependent cyst activation (CDCA) signal is repressed by ciliary Polycystins, cellular Polycystins, and ciliary FPC. FPC cleavage produces $ICD_{15}$. $ICD_{15}$ translocates to mitochondria and produces $ICD_{12}$, which inhibits the propagation of the CDCA signal. The C-terminal tail (CTT) of PC1 may also enter mitochondria and inhibit the propagation of CDCA. These factors lead to the regulation of normal tubule diameter. **b** In $Pkd1^{V/V}$ kidney, $PC1^V$ exhibits impaired localization to the cilia[22,23]. The CDCA is activated but its activity is suppressed by $PC1^V$ in the cell body and by both its CTT and $ICD_{15}$ in mitochondria,

leading to modest cystogenesis. **c** In $\Delta CT/V$ ($Pkhd1^{\Delta67/\Delta67};Pkd1^{V/V}$) kidney, in addition to the activation of CDCA by loss of ciliary PC1, $ICD_{15}$ and $ICD_{12}$ cannot be produced and cannot inhibit the propagation of the CDCA signal, resulting in enhanced cystogenesis compared with $Pkd1^{V/V}$ single mutants. However, FPC-ΔCT is still able to localize to the cilium and reduce CDCA. **d** In $KO/V$ ($Pkhd1^{-/-};Pkd1^{V/V}$) kidney, the only remaining inhibition to the CDCA comes from extra-ciliary $PC1^V$ and possibly its CTT in mitochondria; thus cystogenesis is severe. **e** Summary of the model. FPC and PC1 work together to prevent the initiation and propagation of the cystogenic signal generated in cilia. Figure created in BioRender.

**Commercial primary antibodies.** Mitochondria, Pyruvate Dehydrogenase (Abcam, 13G2AE2BH5; Immunoblot at 1 µg/ml); Aminopeptidase N (Abcam, ab108310; Immunoblot, 1:1000–10,000 dilution); THP, Tamm-Horsfall glycoprotein/Uromodulin (Santa Cruz, sc-20631; Immunoblot, 1:1000 dilution); Aquaporin2, AQP2, (Sigma, A7310; Immunoblot, 1:200–1:1000 dilution); β-Actin (Sigma, A5441; Immunoblot, 1:5000–10,000 dilution); E-cadherin, E-Cad (Cell Signaling, 3195 P; Immunoblot, 1:1000 dilution); TOM20 (Cell Signaling, 42406 T; Immunoblot, 1:500–1000 dilution); Tubulin (Sigma, T6793; Immunoblot, 1:2000 dilution), Actin-Rhodamine (BioRad, 12004163; Immunoblot, 1:1000–10,000 dilution).

**Secondary antibodies.** Goat anti-Rat HRP (Sigma, NA935V); Alexa Fluor donkey anti-mouse 555 (Invitrogen, A31570); Alexa Fluor donkey anti-rabbit 647 (Invitrogen, A31573).

**Western blot**
Cells and kidneys were lysed in either 1% Triton lysis buffer or RIPA buffer with Complete protease inhibitor cocktail added (Roche, 11697498001). Lysates were loaded on 4–12% Bis-Tris protein gels (Invitrogen, NP0336BOX) and run with MES running buffer (Invitrogen, NP0002). Proteins were transferred to polyvinylidene fluoride (PVDF) membranes, blocked with 5% non-fat milk block for 30 min, and incubated with primary antibodies at 4 °C overnight. Membranes were washed with Tris-buffered saline containing Tween (TBST) and probed with fluorescent- or horseradish peroxidase (HRP)-conjugated secondary antibodies prior to detection using either a BioRad ChemiDoc Imaging System or radiographic film.

**Immunohistochemistry**
Tissues were harvested, fixed with 4% paraformaldehyde (PFA), and embedded in paraffin before being sectioned, and stained with hematoxylin and eosin using standard protocols. Lectin staining was carried out on 5 µm paraffin sections. After deparaffinization with xylenes, sections were rehydrated in an ethanol series before boiling in citrate buffer (DAKO Cytomation, S2369) for antigen retrieval. After blocking, sections were incubated with Lotus Tetragonolobus Lectin (LTL)-Fluorescein (Vector Laboratories, FL-1321; used at 2 mg active conjugate/ml) and Dolichos Biflorus Agglutinin (DBA)-

Rhodamine (Vector Laboratories, RL-1032; used at 2 mg active conjugate/ml), for 1 h. After PBS washes, sections were incubated with DAPI, PBS washed, and mounted with Fluoromount-G (Invitrogen, 00-4958-02).

**Cell culture, transfection, and immunofluorescence**
mIMCD3 (ATCC; CRL-2123), MDCK[23] and HEK293 (ATCC, CRL-1573) cells were cultured in DMEM/F12 medium with GlutaMAX (Thermo Fisher Scientific, #31331093), supplemented with 10% FBS, 1% Penicillin-Streptomycin (PenStrep) (Thermo Fisher Scientific, #15070-063) and 1% Sodium Pyruvate (Thermo Fisher Scientific, #11360-039). Transfection was performed using standard lipofectamine2000 (Invitrogen, 11668019) transfection protocols and as previously described[23]. Fluorescent images were obtained using Zeiss Axio Observer D1 microscope and the associated software ZEN (blue edition).

**Split GFP complementation assay**
For the split GFP complementation assay and $ICD_{15}$-GFP localization, mIMCD3 cells were co-transfected with complementary constructs or single construct controls and plated on coverslips in 6-well plates. Growth medium was replaced after 24 h, and cells were allowed to recover for 24 h before fixation with room temperature 4% PFA (10 min). Fixed cells were permeabilized with 0.25% Triton (3 min) and stained with DAPI before mounting with Fluoromount-G (Invitrogen 00-4958-02). For the split GFP assay[48], three separate experiments per construct were imaged 24–48 h after transfection. All constructs were co-transfected with mts-mCherry-$GFP_{1-10}$. Therefore, mitochondria were labeled with mCherry.

**Cellular fractionation**
A serial centrifugation protocol was used to enrich for mitochondria[65]. Briefly, MDCK cell lines stably expressing FPC constructs were scraped in phosphate-buffered saline (PBS) and pelleted by centrifugation. After resuspension in STE medium (250 mM sucrose, 5 mM Tris, 2 mM EGTA; pH 7.4 at 4 °C), cells were homogenized using a Dounce homogenizer. Mitochondria were isolated by differential centrifugation. The final mitochondrial fraction was collected and resuspended in a minimal volume of STE.

### CRISPR/Cas9 generation of *Pkhd1* KO mIMCD3

To generate *Pkhd1* KO mIMCD3, U6gRNA-Cas9-2A-GFP plasmids (Sigma-Aldrich) carrying three distinct custom-designed guide RNA (gRNA) sequences targeting exons 4, 16 and 18 were used (gRNA#1: GACGTCTCTCCGGCCTTCG; gRNA#2: TTGACTCTTGGGGAGCAGAT; gRNA#3: TGCAATCTGGCACCGTTTT). The most efficient guides (gRNA#1 and #2) were employed. Cells were plated on 150 mm$^2$ plates the day before the transfection. Transfection was performed using Lipofectamine 3000 (Thermo Fisher Scientific, #L3000015) following the manufacturer's instructions. 5 µg of plasmid DNA per dish with 1:3 DNA/Lipofectamine ratio were used. The CMV-Cas9-2A-RFP scrambled gRNA was used as a control. Three days after transfection cells were sorted by FACS for GFP (potential *Pkhd1* KO mIMCD3) or RFP (control mIMCD3) and plated as single cells into 96-wells plates. The vital clones were sequentially expanded and screened for FPC protein expression by western blot. Four out of nine *Pkhd1* KO clones were established by guides #1 or #2. As controls, nine RFP clones were generated (Cassina and Boletta, in preparation). The guide RNA sequence targeting exon 2 of murine *Pkhd1* sequence (gRNA#4: 5′-ACTCCCTGGAAATGCGCTCTGG-3′) was also successfully used in the system.

### Seahorse metabolic flux analysis

The day before the assay, 15,000 cells per well in 96-well Seahorse cell culture and incubated in a 5% $CO_2$ incubator at 37 °C overnight. The day after the culture medium was changed with Seahorse XF DMEM medium (Agilent Technologies, #103575-100) supplemented with 10 mM glucose, 1 mM sodium pyruvate, and 2 mM L-glutamine. The plate was incubated at 37 °C for 1 h in a non-CO2 incubator before starting the assay. We performed Mito Stress Test assay. After OCR baseline measurements oligomycin A (O), carbonyl cyanide 4-(trifluoromethoxy) phenylhydrazone (FCCP), and antimycin A/rotenone (A/R) were added sequentially to each well, to reach the final concentrations of 1 µM O, 1.5 µM FCCP, and 0.5 µM A/R. Results were normalized by cell number using CyQUANT Cell Proliferation Assays (Thermo Fisher Scientific, #C35011). Results are mean ± SD of technical replicates, OCR data are expressed as pmol of oxygen per minute per arbitrary units (pmol/min/a.u.). All the analyses were performed with the Agilent Seahorse Wave software (Agilent). Basal respiration was calculated by subtracting the minimum OCR measurement after A/R injection from the last OCR measurement before oligomycin injection, maximal respiration by subtracting the last OCR measurement before oligomycin injection from the maximum rate measured after FCCP injection, spare respiratory capacity by subtracting the basal respiration from the maximal respiration.

### *N*-glycosylation analysis

Samples were denatured using glycoprotein denature buffer (New England Biolabs) for 1 min at 95 °C and then quickly chilled on ice. The denatured glycoprotein was incubated with PNGaseF or EndoH (New England Biolabs) for 1 h at 37 °C.

### Kidney dissection and tubules isolation

Kidneys were harvested under sterile conditions from P6 WT and *Pkhd1* null mice, followed by mincing the kidneys into 1 mm$^3$ cubes and incubating in collagenase II digestion (0.1 g/ml, Sigma C2139) solution at 37 °C with shaking for 30 mins. Tubule cells were collected by centrifugation and resuspended in DMEM/F12 containing DNase I (2U/ml, Sigma D4263). After 3 mins of DNase I digestion, tubule cells were collected by centrifugation and the resuspended pellet was filtered through 70 µm pore filters to isolate tubules and exclude glomeruli. The tubules were washed in ice-cold PBS and pulsed in a centrifuge three times to remove collagenase and red blood cells. After each spin, the pellets become increasingly opaque white. The remaining pellets are isolated kidney tubule fragments.

### Transmission electron microscopy

P6 kidneys were collected and fixed using TEM fixative (2% Paraformaldehyde, 2.5% Glutaraldehyde, 2 mM $CaCl_2$ in 0.1 M PIPES buffer, pH 7.35) and postfixed with reduced osmium. Fixed specimens were washed with buffer, dehydrated in a graded series of ethanols, embedded in plastic resin, and cut to 80–100 nm sections by *Leica UC6* ultramicrotome. The sections were stained with uranyl acetate and Sato's triple lead stain, and images were collected of whole tubules, followed by higher magnification images of individual cells and mitochondria by transmission electron microscope FEI tecnai T12 at the Electron Microscopy Core Imaging Facility of the University of Maryland Baltimore.

### Analysis and Statistics

Mitochondrial localization signal prediction was undertaken using Mitoprot II[46] (https://ihg.helmholtz-muenchen.de/ihg/mitoprot.html). Mitochondria shape descriptors were calculated using ImageJ software (http://rsb.info.nih.gov/ij/) (version 1.54 f). Averages are represented in graphs with SEM error bars. Images were blinded to genotype and processed for shape descriptor analysis and cristae diameter measurements. Data analysis was undertaken using Microsoft Excel (office 365), R studio (version 1.4.1106) (https://www.rstudio.com/).

### Cystic area calculation

Images of sections of P0 kidneys were analyzed using ImageJ software. Kidney slices were binarized, and cyst areas were measured using "analyze particles". All fenestrations >10 µm diameter were counted. The "cyst" areas were compared with the area of the kidney slice to produce a % cystic area.

### Reporting summary

Further information on research design is available in the Nature Portfolio Reporting Summary linked to this article.

## Data availability

All data generated or analyzed during this study are included in this article (and its supplementary information files). Source data are provided with this paper.

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

## Acknowledgements

This work was supported by National Institute of Health (NIH) grants R01 DK111611 and R01 DK125404 (FQ), Polycystic Kidney Disease Foundation grant 236G19a (FQ), the University of Maryland Baltimore Institute for Clinical & Translational Research grant (Project ID 14) (FQ), a Polycystic Kidney Disease Foundation fellowship grant (215F19a) (RW) and by funds from the Italian Association for Research on PKD (A.B.). We also acknowledge the Polycystic Kidney Disease Research Resource Consortium (U54DK126114) for providing us with the E1, E3, and E4 antibodies used in this study. We thank Gianfranco Distefano for technical help with the generation of mIMCD3 cells KO for *Pkhd1*. We thank Gary Fiskum for his critical reading of the manuscript and helpful discussions, and Ru-Ching Hsia and Christine Brantner for assistance in transmission electron microscopy.

## Author contributions

F.Q., R.V.W. and Q.Y. developed the concept of this study. R.V.W., H.X., Q.Y., A.M., B.M.P., K.F.S., S.R., R.L., L.C., A.B. and F.Q. designed and performed experiments, and analyzed data. P.O. and T.W. provided reagents. R.V.W. and F.Q. prepared the draft and final version of the manuscript. All authors read and approved the final manuscript.

## Competing interests

The authors have no competing interests to declare.
