## [Peer Review File · Nature Communications]

Fibrocystin/Polyductin releases a C-terminal fragment that translocates into mitochondria and suppresses cystogenesisREVIEWER COMMENTS

Reviewer #1 (Remarks to the Author):

This study presents an interesting story, which try to state that C-terminal fragments of FPC, ~15 kDa (ICD15), ~12 kDa (ICD12), and ~6 kDa (ICD6), translocate into mitochondria and have a prevent function from cystogenesis. Better support should be provided for the key conclusions “prevent cystogenesis” by the C-terminal fragments. There are some questions:

1. PKHD1 has different isoforms. Cleaved fragments of FPC was found in this study. Were these proteins sequenced or, at part, has a same amino acid peptide as encoded in exon 67?
2. The authors found ICD15 translocated to mitochondria. Was mitochondrial biogenesis activated or mitochondrial structure or function altered when ICD15 translocated?
3. In the text, mitochondrial ultrastructure altered in Pkhd1 KO mice, compared to the Wt controls. To evaluate the ICD15 function or its impact on mitochondria, it is better to provide additional data of mitochondrial structure and function in vitro, compared with Pkhd1 KO, ICD15 transfected back, and Wt.
4. In line 375 “Both compound heterozygous Pkhd1 Δ 3-4/+;Pkd1V/+ mutant mice....” . Compound heterozygote is referred to two different mutations (or variants) in one same gene. As Pkhd1 and Pkd1 are two different genes, it should correct to be “digenic mutant mice”.
5. The 5th result section (line 429-468) probably be better as a part of one result section or as a supplementary data.

Others:

6. the cited refs37 in line138 and refs19 in line 354 should be confirmed (probably it should be corrected to 38 or 20 respectively).
7. line 342-343, sentence “This result suggests that FPC may participate in regulating mitochondrial structure and function”. However, no mitochondrial function has been provided in this section.
8. Sentence in line 373-375, “The Pkhd1 Δ 3-4 allele has a deletion of exons 3-4.....in the

extracellular domain” is better to put in the place when Pkhd1 Δ 3-4 first appeared.

9. In Figure 2, C) D) E) F) I) graphs should clearly show the distribution of data and variation (as dot plot in the column), and the number of times of the experiments should be stated in figure legends.

Reviewer #2 (Remarks to the Author):

The paper lacks sufficient novelty. All single mutant phenotypes were previously described. The genetic interaction between Pkd1 and Pkhd1 (double mutant phenotypes) was previously described by several groups, including this group. Pkhd1/FPC deficiency and mitochondrial defects were previously described. Cleavage of Pkhd1 protein FPC has been reported previously by other group and this group. This paper described a more detailed analysis of FPC CTT responsible for mitochondrial translocation. They found in Pkd1 and Pkhd1 double mutants develop embryonic pancreatic cysts that is present in Pkd1 null mutant. This finding is of some interest but of limited significance. The paper is too long: Introduction, 4 pages; Results, 12 pages; and Discussion is 6 pages, difficult to read. Nomenclature is confusing. What is Pkhd1 $^{-/-}$, Pkhd1 Δ 3-4, Pkhd1 Δ CT, Pkhd1 Δ 67, KO. There are 2 Pkd1 lines (Pkd1RC, Pkd1V) and 5 Pkhd1 alleles used in the study or some of them are the same thing? In the methods, there is almost no information about all the mouse lines used in the study. The conclusion is not fully supported by the experimental data.

Reviewer #3 (Remarks to the Author):

The authors describe that the type I transmembrane protein Fibrocystin/Polyductin (FPC), the main protein for autosomal recessive polycystic kidney disease (ARPKD) encoded by the large PKHD1 gene, undergoes complex proteolytic processing in vivo. This processing results in three small soluble C-terminal fragments with molecular weights ranging from 6-15 kDa (termed ICD6, 12, and 15). At the N-terminus of ICD15 the authors identified a mitochondrial targeting sequence (MTS) that directs its mitochondrial localization. Walker et al. demonstrate convincing ultrastructural alterations of mitochondria of Pkhd1 KO mouse kidney tubules in contrast to wildtype controls.

In addition, the authors performed comprehensive genetic epistasis studies in mice and could show that FPC inactivation aggravates renal cystogenesis in a Pkd1 mouse mutant (Pkd1v/v). Deletion of the final exon of Pkhd1 (exon 67) enhanced cystogenesis in the kidney, but not in the pancreas, of those Pkd1v/v mutant mice. Overall, the novel data generated by this study indicates that an important function of the ICD protein products within the mitochondria is to protect against cyst formation when the function of Polycystin-1, the Pkd1 gene product and major ADPKD protein, is compromised.

This is an excellent study that adds important knowledge to the pathophysiology of polycystic kidney disease. I have only minor comments that the authors are recommended to address:

- A substantial amount of data has been generated by the authors, however, the manuscript and its readability would clearly benefit from some kind of shortening (e. g. in the introduction section).

- Page 3: Please add the information that (while PKHD1 clearly defines the main ARPKD gene) pathogenic variants in various other genes may cause ARPKD or ARPKD-like phenotypes.

RESPONSE TO REFEREES

We thank reviewers 1 and 3 for accepting the revised manuscript in its current form. Below, we present point-by-point responses to reviewer 2.

Reviewer #2 (Remarks to the Author)

In this revised manuscript, the authors have improved the clarity of nomenclature and a table describing the mouse lines used in this study is helpful. However, several major points still warrant attention.

“3) By utilizing the Pkd1V/V model in the genetic epistasis analyses, we have made the intriguing discovery that loss of FPC leads to a Pkd1 null-like cystic phenotype both in the kidney and pancreas. Moreover, unlike Pkd1RC used in Olson et al study, Pkd1V/V displayed impaired localization of PC1 to cilia, suggesting specific involvement of cilia in cyst progression as discussed and presented in our model (Figure 7).”

The Pkd1 RC mutant was previously identified as having a maturation defect in PC1, leading to compromised cleavage, as documented by Hopp et al in 2012. Furthermore, research by Su et al in 2015 showed that the RC mutation hinders the proper localization of PC1 to cilia. Consequently, even though the Pkd1 V mutant carries a distinct genetic mutation compared to the Pkd1 RC mutant, which was formerly employed in Pkhd1-Pkd1 digenic study by Olsen et al, the observed distinction remains marginal, hence the novelty in this regard is inadequate.

Response:

We appreciate the reviewer's recognition of the improved clarity in nomenclature in the revised manuscript. While we acknowledge the reviewer's concern regarding the distinction between the Pkhd1-Pkd1 interaction using the Pkd1V/V mutant versus the Pkd1 RC mutant, it's important to highlight that our study's main focus goes far beyond just the Pkhd1-Pkd1 interaction.

The true novelty lies in the new insights derived from the utilization of the Pkd1V/V strain. As we outlined extensively in our prior response, our study introduces multiple noteworthy findings. These encompass the translocation of ICD15 into mitochondria, the identification of ultrastructural and functional mitochondrial abnormalities in Pkhd1 KO kidney tubules and renal epithelial cells, and the essential role of ICD15 in suppressing renal cyst development in mice. Consequently, our manuscript's contribution goes significantly beyond this single observation, offering novel perspectives on FPC's role in ARPKD and signifying a substantial advancement in the field.

We had thoroughly discussed the rationale and advantages of utilizing the Pkd1V/V strain, particularly for assessing proximal tubules, in lines 358-369, as provided in the previous revision. The results obtained through our utilization of the Pkd1V/V mutants offer valuable novel insights into the role of FPC in ARPKD and constitute a significant contribution to the current body of literature.

“Our discovery sheds light on a distinct mechanism of the ciliary function of FPC in pancreatic cystogenesis”

Can the authors explain based on what evidence they connected the occurrence of pancreatic cysts with the ciliary function of FPC?

Response:

Our identification of pancreatic cysts in the two digenic mutants ($\Delta 3-4/V$ and KO/V) provides compelling evidence for the pivotal role of FPC in cystogenesis within the pancreas. Notably, our analysis of $\Delta 67/V$ digenic mutants revealed an intriguing pattern: while renal cystogenesis in $\Delta 67/V$ was enhanced compared to Pkd1V/V single mutants, no pancreatic cysts were observed. These results suggest that while FPC plays a crucial role in cystogenesis in the pancreas, its ICD15 component does not exert a substantial impact on this process when combined with the Pkd1V/V mutation.

We have comprehensively discussed this intriguing finding in the revised manuscript in lines 583-597 regarding the ciliary function of FPC in safeguarding the pancreas from cystogenesis through mechanisms extending beyond its C-terminal tail. This notion is based on the observation (shown in Figure 1G) that Pkhd1 $\Delta 67/\Delta 67$ expresses the FPC- Δ CT protein. This variant lacks ICD15 but retains the intact ciliary targeting sequence, suggesting its likely localization to the cilia. Consequently, we hypothesize that this ciliary function could **protect** the pancreas from cystogenesis through the involvement of its ecto- and transmembrane domains.

“We propose a molecular mechanism to explain the interaction between Pkhd1 and Pkd1 in cystogenesis through a cilia-mitochondria connection (Figure 7).”

Please specify the experimental evidence that supports this model. Which experiments and figures are pertinent to establishing the cilia-mitochondria connection?

Response:

The proposed model is based on a synthesis of prior research and the results of this study. In our text, we reference prior studies that introduced the concept of the cilia-dependent cyst activation (CDCA) mechanism, which triggers cystogenesis upon the depletion of polycystins—integral ciliary membrane proteins, as outlined in lines 644-649.

Our model logically bridges the previously established ciliary roles of FPC and polycystins with FPC's mitochondrial function through ICD15 as revealed by our current study. The connection between cilia and mitochondria is explicitly articulated in the initial segment of the Discussion (lines 533-535), following a concise overview of the pivotal discoveries of our study. Further elaboration on the model depicted in Figure 7 is provided in the concluding section of the Discussion.

In our preceding round of revisions, we made extensive efforts to streamline the content, enhance clarity, and improve the manuscript's overall readability. The presentation logically follows from the progression of our presentation and is expounded upon in the preceding sections of the Discussion. Consequently, we have chosen to refrain from introducing supplementary elaboration or referencing specific experiments or figures in this section. Such additions might introduce redundancy and unnecessarily prolong the text—a situation we are keen to circumvent based on prior feedback.

To amplify clarity, we have now added references to the pertinent figures (indicated in blue) that correspond to each of the key findings summarized in the initial section of the Discussion (lines 520-532).

Olsen et al conducted RNA-seq analysis on digenic and monogenic mouse models, highlighting the ciliary compartment as a commonly dysregulated target. This was underscored by enhanced ciliary expression and altered length in the digenic model. However, they did not identify any defects in polycystin-2 ciliary localization within their Pkhd1 KO mice.

Response:

We acknowledge the alignment of our findings with those of Olsen et al., as noted in the manuscript (lines 133-138). However, it is important to clarify that our study did not investigate the role of polycystin-2. Therefore, the notion of the reviewer “**However, they did not identify any defects in polycystin-2 ciliary localization within their Pkhd1 KO mice**” is not directly applicable to our study.

Are mitochondrial defects the sole abnormalities discovered in the Pkhd1 KO mice in the present study? Have any other ciliary defects been detected? Is PC1 ciliary localization normal across different Pkhd1 KO lines? Additionally, what impact does FPC KO have on PC1 C-tail cleavage?

Notably, a decrease in mitochondrial mass and a PKD1-dose-dependent decline in mitochondrial membrane potential have been previously observed in renal tubular cells. Existing studies demonstrate that cleaved PC1 C-tail induces mitochondrial dysfunction, and mitochondrial abnormalities are evident in Pkd1 knockout mice. Have the authors investigated mitochondrial function in the digenic animals as well? This should be done.

Response:

We appreciate the reviewer's thought-provoking new inquiries, which were not brought up in the previous critique. In our present study, we did not assert that mitochondrial defects are the sole abnormalities observed in Pkhd1 KO mice. Additionally, we recognize that inquiries, such as those regarding the cleaved PC1 C-tail, remain unexplored. However, we want to highlight that we did address the potential significance of FPC on the cleaved PC1 C-tail in our previous revision, as outlined in lines 564-571.

We concur that the questions related to the digenic animals indeed merit attention. Nevertheless, we view them as potential avenues for future investigations.

In this study, the authors have identified mitochondrial defects in the kidneys of Pkhd1 knockout mice, despite the absence of cyst manifestation. The functional significance of these mitochondria defects in Pkhd1 KO mice raises pertinent questions. Does the lack of renal cysts in Pkhd1 KO mice suggest that mitochondrial abnormalities alone are insufficient to trigger cystogenesis? Alternatively, could it imply that the extent of mitochondrial defects in Pkhd1 mice is not substantial enough to drive cyst formation? Earlier research, notably the work of Nishio et al in 2010, has firmly established the absence of oriented cell division as a distinctive feature of Pkhd1 mutations.

Response:

We appreciate the reviewer's recognition of our findings pertaining to the significance of mitochondrial defects in the kidneys of Pkhd1 knockout mice, even in the absence of observable cyst manifestation. This finding indeed stands as a noteworthy and innovative aspect of our study. It is indisputable that a multitude of dysregulated pathways contribute to a complex process such as cystogenesis, including a defect in oriented cell division. Our study raises important questions about whether mitochondrial abnormalities alone are insufficient to trigger cystogenesis or if other factors come into play, warranting further investigations. In its entirety, we believe that our study will make a significant and valuable addition to the existing literature.

Overall, the mechanisms underlying the genetic epistasis analyses conducted in this

study remain unclear. The model in Figure 7 is too hypothetical, which is not supported by data.

Response:

We concur that although our study unveils a novel and critical mitochondrial role of FPC, a prominent cystoprotein, in cystogenesis, the precise mechanisms underlying the genetic epistasis analyses conducted in this study remain enigmatic. It is indeed a humbling realization that a study of this nature cannot comprehensively delve into the intricacies of the next-level mechanisms.

Nonetheless, it's important to highlight that our current study has provided valuable insights into renal cystogenesis. These insights, while not exhaustive, have the potential to stimulate further exploration of these intricate questions by the research community. We therefore believe that our study would make a significant and valuable addition to the existing literature.

While we acknowledge that our model in Figure 7 is hypothetical, as models inherently are, we respectfully disagree with the characterization of its level of hypothesis as being "TOO hypothetical." As previously discussed, the model is consistent with prior research and provides a coherent explanation for our current findings.

REVIEWERS' COMMENTS

Reviewer #1 (Remarks to the Author):

I am happy that the authors have now addressed the concerns from the previous review.

Reviewer #2 (Remarks to the Author):

In this revised manuscript, the authors have improved the clarity of nomenclature and a table describing the mouse lines used in this study is helpful. However, several major points still warrant attention.

“3) By utilizing the Pkd1V/V model in the genetic epistasis analyses, we have made the intriguing discovery that loss of FPC leads to a Pkd1 null-like cystic phenotype both in the kidney and pancreas. Moreover, unlike Pkd1RC used in Olson et al study, Pkd1V/V displayed impaired localization of PC1 to cilia, suggesting specific involvement of cilia in cyst progression as discussed and presented in our model (Figure 7).”

The Pkd1 RC mutant was previously identified as having a maturation defect in PC1, leading to compromised cleavage, as documented by Hopp et al in 2012. Furthermore, research by Su et al in 2015 showed that the RC mutation hinders the proper localization of PC1 to cilia. Consequently, even though the Pkd1 V mutant carries a distinct genetic mutation compared to the Pkd1 RC mutant, which was formerly employed in Pkhd1-Pkd1 digenic study by Olsen et al, the observed distinction remains marginal, hence the novelty in this regard is inadequate.

“Our discovery sheds light on a distinct mechanism of the ciliary function of FPC in pancreatic cystogenesis”

Can the authors explain based on what evidence they connected the occurrence of pancreatic cysts with the ciliary function of FPC?

“We propose a molecular mechanism to explain the interaction between Pkhd1 and Pkd1 in cystogenesis through a cilia-mitochondria connection (Figure 7).”

Please specify the experimental evidence that supports this model. Which experiments and figures are pertinent to establishing the cilia-mitochondria connection?

Olsen et al conducted RNA-seq analysis on digenic and monogenic mouse models, highlighting the ciliary compartment as a commonly dysregulated target. This was underscored by enhanced ciliary expression and altered length in the digenic model. However, they did not identify any defects in polycystin-2 ciliary localization within their Pkhd1 KO mice.

Are mitochondrial defects the sole abnormalities discovered in the Pkhd1 KO mice in the present study? Have any other ciliary defects been detected? Is PC1 ciliary localization normal across different Pkhd1 KO lines? Additionally, what impact does FPC KO have on PC1 C-tail cleavage?

Notably, a decrease in mitochondrial mass and a PKD1-dose-dependent decline in mitochondrial membrane potential have been previously observed in renal tubular cells. Existing studies demonstrate that cleaved PC1 C-tail induces mitochondrial dysfunction, and mitochondrial abnormalities are evident in Pkd1 knockout mice. Have the authors investigated mitochondrial function in the digenic animals as well? This should be done.

In this study, the authors have identified mitochondrial defects in the kidneys of Pkhd1 knockout mice, despite the absence of cyst manifestation. The functional significance of these mitochondria defects in Pkhd1 KO mice raises pertinent questions. Does the lack of renal cysts in Pkhd1 KO mice suggest that mitochondrial abnormalities alone are insufficient to trigger cystogenesis? Alternatively, could it imply that the extent of mitochondrial defects in Pkhd1 mice is not substantial enough to drive cyst formation?

Earlier research, notably the work of Nishio et al in 2010, has firmly established the absence of oriented cell division as a distinctive feature of Pkhd1 mutations.

Overall, the mechanisms underlying the genetic epistasis analyses conducted in this study remain unclear. The model in Figure 7 is too hypothetical, which is not supported by data.

Reviewer #3 (Remarks to the Author):

Congratulations to this wonderful study, I have no further comments.

RESPONSE TO REFEREES

We thank reviewers 1 and 3 for accepting the revised manuscript in its current form. Below, we present point-by-point responses to reviewer 2 for the final version (NCOMMS-22-35324B).

Reviewer #2 (Remarks to the Author)

In this revised manuscript, the authors have improved the clarity of nomenclature and a table describing the mouse lines used in this study is helpful. However, several major points still warrant attention.

“3) By utilizing the Pkd1V/V model in the genetic epistasis analyses, we have made the intriguing discovery that loss of FPC leads to a Pkd1 null-like cystic phenotype both in the kidney and pancreas. Moreover, unlike Pkd1RC used in Olson et al study, Pkd1V/V displayed impaired localization of PC1 to cilia, suggesting specific involvement of cilia in cyst progression as discussed and presented in our model (Figure 7).”

The Pkd1 RC mutant was previously identified as having a maturation defect in PC1, leading to compromised cleavage, as documented by Hopp et al in 2012. Furthermore, research by Su et al in 2015 showed that the RC mutation hinders the proper localization of PC1 to cilia. Consequently, even though the Pkd1 V mutant carries a distinct genetic mutation compared to the Pkd1 RC mutant, which was formerly employed in Pkhd1-Pkd1 digenic study by Olsen et al, the observed distinction remains marginal, hence the novelty in this regard is inadequate.

Response:

We appreciate the reviewer's recognition of the improved clarity in nomenclature in the revised manuscript. While we acknowledge the reviewer's concern regarding the distinction between the Pkhd1-Pkd1 interaction using the Pkd1V/V mutant versus the Pkd1 RC mutant, it's important to highlight that our study's main focus goes far beyond just the Pkhd1-Pkd1 interaction.

The true novelty lies in the new insights derived from the utilization of the Pkd1V/V strain. As we outlined extensively in our prior response, our study introduces multiple noteworthy findings. These encompass the translocation of ICD15 into mitochondria, the identification of ultrastructural and functional mitochondrial abnormalities in Pkhd1 KO kidney tubules and renal epithelial cells, and the essential role of ICD15 in suppressing renal cyst development in mice. Consequently, our manuscript's contribution goes significantly beyond this single observation, offering novel perspectives on FPC's role in ARPKD and signifying a substantial advancement in the field.

We had thoroughly discussed the rationale and advantages of utilizing the Pkd1V/V strain, particularly for assessing proximal tubules, in lines 354-365 (of the final version), as provided in the previous revision. The results obtained through our utilization of the Pkd1V/V mutants offer valuable novel insights into the role of FPC in ARPKD and constitute a significant contribution to the current body of literature.

“Our discovery sheds light on a distinct mechanism of the ciliary function of FPC in pancreatic cystogenesis”

Can the authors explain based on what evidence they connected the occurrence of pancreatic cysts with the ciliary function of FPC?

Response:

Our identification of pancreatic cysts in the two digenic mutants ($\Delta 3-4/V$ and KO/V) provides compelling evidence for the pivotal role of FPC in cystogenesis within the pancreas. Notably, our analysis of $\Delta 67/V$ digenic mutants revealed an intriguing pattern: while renal cystogenesis in $\Delta 67/V$ was enhanced compared to Pkd1V/V single mutants, no pancreatic cysts were observed. These results suggest that while FPC plays a crucial role in cystogenesis in the pancreas, its ICD15 component does not exert a substantial impact on this process when combined with the Pkd1V/V mutation.

We have comprehensively discussed this intriguing finding in the revised manuscript in lines 578-592 regarding the ciliary function of FPC in safeguarding the pancreas from cystogenesis through mechanisms extending beyond its C-terminal tail. This notion is based on the observation (shown in Figure 1g) that Pkhd1 $\Delta 67/\Delta 67$ expresses the FPC- Δ CT protein. This variant lacks ICD15 but retains the intact ciliary targeting sequence, suggesting its likely localization to the cilia. Consequently, we hypothesize that this ciliary function could **protect** the pancreas from cystogenesis through the involvement of its ecto- and transmembrane domains.

“We propose a molecular mechanism to explain the interaction between Pkhd1 and Pkd1 in cystogenesis through a cilia-mitochondria connection (Figure 7).”

Please specify the experimental evidence that supports this model. Which experiments and figures are pertinent to establishing the cilia-mitochondria connection?

Response:

The proposed model is based on a synthesis of prior research and the results of this study. In our text, we reference prior studies that introduced the concept of the cilia-dependent cyst activation (CDCA) mechanism, which triggers cystogenesis upon the depletion of polycystins—integral ciliary membrane proteins, as outlined in lines 639-644.

Our model logically bridges the previously established ciliary roles of FPC and polycystins with FPC's mitochondrial function through ICD15 as revealed by our current study. The connection between cilia and mitochondria is explicitly articulated in the initial segment of the Discussion (lines 528-531), following a concise overview of the pivotal discoveries of our study. Further elaboration on the model depicted in Figure 7 is provided in the concluding section of the Discussion.

In our preceding round of revisions, we made extensive efforts to streamline the content, enhance clarity, and improve the manuscript's overall readability. The presentation logically follows from the progression of our presentation and is expounded upon in the preceding sections of the Discussion. Consequently, we have chosen to refrain from introducing supplementary elaboration or referencing specific experiments or figures in this section. Such additions might introduce redundancy and unnecessarily prolong the text—a situation we are keen to circumvent based on prior feedback.

To amplify clarity, we have now added references to the pertinent figures that correspond to each of the key findings summarized in the initial section of the Discussion (lines 515-531).

Olsen et al conducted RNA-seq analysis on digenic and monogenic mouse models, highlighting the ciliary compartment as a commonly dysregulated target. This was underscored by enhanced ciliary expression and altered length in the digenic model. However, they did not identify any defects in polycystin-2 ciliary localization within their Pkhd1 KO mice.

Response:

We acknowledge the alignment of our findings with those of Olsen et al., as noted in the manuscript (lines 131-136). However, it is important to clarify that our study did not investigate the role of polycystin-2. Therefore, the notion of the reviewer “**However, they did not identify any defects in polycystin-2 ciliary localization within their Pkhd1 KO mice** “ is not directly applicable to our study.

Are mitochondrial defects the sole abnormalities discovered in the Pkhd1 KO mice in the present study? Have any other ciliary defects been detected? Is PC1 ciliary localization normal across different Pkhd1 KO lines? Additionally, what impact does FPC KO have on PC1 C-tail cleavage?

Notably, a decrease in mitochondrial mass and a PKD1-dose-dependent decline in mitochondrial membrane potential have been previously observed in renal tubular cells. Existing studies demonstrate that cleaved PC1 C-tail induces mitochondrial dysfunction, and mitochondrial abnormalities are evident in Pkd1 knockout mice. Have the authors investigated mitochondrial function in the digenic animals as well? This should be done.

Response:

We appreciate the reviewer's thought-provoking new inquiries, which were not brought up in the previous critique. In our present study, we did not assert that mitochondrial defects are the sole abnormalities observed in Pkhd1 KO mice. Additionally, we recognize that inquiries, such as those regarding the cleaved PC1 C-tail, remain unexplored. However, we want to highlight that we did address the potential significance of FPC on the cleaved PC1 C-tail in our previous revision, as outlined in lines 559-566.

We concur that the questions related to the digenic animals indeed merit attention. Nevertheless, we view them as potential avenues for future investigations.

In this study, the authors have identified mitochondrial defects in the kidneys of Pkhd1 knockout mice, despite the absence of cyst manifestation. The functional significance of these mitochondria defects in Pkhd1 KO mice raises pertinent questions. Does the lack of renal cysts in Pkhd1 KO mice suggest that mitochondrial abnormalities alone are insufficient to trigger cystogenesis? Alternatively, could it imply that the extent of mitochondrial defects in Pkhd1 mice is not substantial enough to drive cyst formation? Earlier research, notably the work of Nishio et al in 2010, has firmly established the absence of oriented cell division as a distinctive feature of Pkhd1 mutations.

Response:

We appreciate the reviewer's recognition of our findings pertaining to the significance of mitochondrial defects in the kidneys of Pkhd1 knockout mice, even in the absence of observable cyst manifestation. This finding indeed stands as a noteworthy and innovative aspect of our study. It is indisputable that a multitude of dysregulated pathways contribute to a complex process such as cystogenesis, including a defect in oriented cell division. Our study raises important questions about whether mitochondrial abnormalities alone are insufficient to trigger cystogenesis or if other factors come into play, warranting further investigations. In its entirety, we believe that our study will make a significant and valuable addition to the existing literature.

Overall, the mechanisms underlying the genetic epistasis analyses conducted in this

study remain unclear. The model in Figure 7 is too hypothetical, which is not supported by data.

Response:

We concur that although our study unveils a novel and critical mitochondrial role of FPC, a prominent cystoprotein, in cystogenesis, the precise mechanisms underlying the genetic epistasis analyses conducted in this study remain enigmatic. It is indeed a humbling realization that a study of this nature cannot comprehensively delve into the intricacies of the next-level mechanisms.

Nonetheless, it's important to highlight that our current study has provided valuable insights into renal cystogenesis. These insights, while not exhaustive, have the potential to stimulate further exploration of these intricate questions by the research community. We therefore believe that our study would make a significant and valuable addition to the existing literature.

While we acknowledge that our model in Figure 7 is hypothetical, as models inherently are, we respectfully disagree with the characterization of its level of hypothesis as being "TOO hypothetical." As previously discussed, the model is consistent with prior research and provides a coherent explanation for our current findings.